

# Tensor network representations
# from the geometry of entangled states

**Matthias Christandl[1], Angelo Lucia[1,2,3], Péter Vrana[1,4,5] and Albert H. Werner[1,2⋆]**

**1** QMATH, Department of Mathematical Sciences, University of Copenhagen,
Universitetsparken 5, 2100 Copenhagen, Denmark
**2** NBIA, Niels Bohr Institute, University of Copenhagen,
Blegdamsvej 17, 2100 Copenhagen, Denmark
**3** Walter Burke Institute for Theoretical Physics and Institute for Quantum Information and
Matter, California Institute of Technology, Pasadena, CA 91125, USA
**4** Department of Geometry, Budapest University of Technology and Economics,
Egry József u. 1., 1111 Budapest, Hungary
**5** MTA-BME Lendület Quantum Information Theory Research Group

⋆ werner@math.ku.dk

## Abstract

Tensor networks provide descriptions of strongly correlated quantum systems based on an underlying entanglement structure given by a graph of entangled states along the edges that identify the indices of the local tensors to be contracted. Considering a more general setting, where entangled states on edges are replaced by multipartite entangled states on faces, allows us to employ the geometric properties of multipartite entanglement in order to obtain representations in terms of superpositions of tensor network states with smaller effective dimension, leading to computational savings.

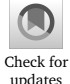
# 1  Introduction

Taming the exponential growth of complexity with increasing system size presents one of the major problems in the theory of quantum many-body systems. Tailor-made Ansatz-classes such as tensor network states have allowed for tremendous progress over the last two decades both in terms of numerical [1–4] as well as analytical work [5,6]. This includes results on ground state properties [7–9], the classification of quantum phases [10,11], disordered systems [12–16], the behaviour of open quantum many-body systems [17,18], critical systems [19], as well as related to the AdS/CFT-correspondence [20].

At the heart of such tensor network approaches is the idea to obtain a class of physical states of interest from an underlying resource state by the application of local linear operations, which can be seen as applying stochastic local operations and classical communication [21]. In the case of matrix product states (MPS) and projected entangled pair states (PEPS) these states are given by networks of maximally entangled states. For certain applications, other tensor network structures have been introduced such as tree tensor networks [22,23] and the multi-scale renormalization ansatz (MERA) [24,25], the latter capturing ground state properties of critical systems.

Another route to generalizing MPS and PEPS, which has been recently explored, allows for more general resource states beyond EPR-pairs [26–28]. These are based on multi-partite quantum states shared among several lattice sites such as GHZ-states [27]. In this work, we further generalize this approach by extending both the underlying resource state or entanglement structure as well as the class of allowed operations. More precisely, we allow for one-parameter families of approximate representations, which reproduce the state of interest to an arbitrary precision.

We show how these approximate representations can be turned into exact representations in terms of a linear superposition of a moderate number of tensor network states. This approach provides more efficient tensor network representations for certain classes of states, and gives rise to an efficient algorithm to reconstruct expectation values faithfully. In addition, we obtain results that allow to simulate or re-express tensor network states based on multi-partite resource states in terms of ordinary PEPS, thereby enabling a numerical treatment of these states by the highly optimized methods that exist for PEPS. As a concrete example, we show that that semi-injective PEPS on the two-dimensional square lattice based on GHZ states as introduced in [27] with bond dimension $D$ can be represented as a normal PEPS of bond dimension $2D$.

As an example of the application of our results, we consider the Resonating Valence Bond (RVB) state, which has originally been proposed as the ground state of spin liquids [29] and is also of importance in the theory of high-temperature superconductivity [30]. The RVB state has also been studied extensively in the context of PEPS [31–33]. A first tensor network representation of this state as a PEPS with bond dimension equal to 3 was introduced in [31]. We present two new representations of the state: a PEPS with non-uniform bond dimensions

on the kagome lattice, with bonds of dimension $(2, 2, 3)$ depending on the orientation of the bond, which we show is optimal; and a representation in terms of a superposition of a linear (in the system size) number of PEPS with bond dimension equal to 2.

The paper is organized as follows. In Section 2, we revisit the definition of MPS and connect it to notions in algebraic complexity theory. In particular, we introduce degenerations as a way of obtaining approximate state representations with smaller bond dimension. This leads to the concept of border bond dimension and we give a first example in terms of the W-state where this approximate representation leads to a provably more efficient representation. These ideas are then generalized in Section 3 to PEPS and other entanglement structures of multi-partite states, seen as representations of graphs and hypergraphs. In Section 4, we consider the question how to transform a given entanglement structure into another one based on degenerations and provide an efficient algorithm to compute exact expectation values even in this approximate setting. At the same time, this result lets us interpret states obtained from degenerations as arising from superpositions of tensor network states with the number of superimposed states growing linearly with the system size.

The main building block for this general result turns out to be an approximate conversion between the plaquette states of the two entanglement structures involved. Therefore, we present in Section 5 specific examples of such plaquette conversions between important tensor network classes such as PEPS, generalized injective PEPS and the RVB state on the kagome lattice, proving lower and upper bounds on the required bond dimensions. In Appendix A we include an analysis of the computational cost of computing expectation values of tensor network states on the square and kagome lattice using these more efficient approximate representations, focusing in particular on the RVB state.

## 2 Matrix product states & Algebraic complexity theory

As a starting point for more general tensor networks, we discuss in this section the concept of MPS representations from the point of view of algebraic complexity theory. In particular, we introduce the concept of degenerations, which correspond to a weaker notion of MPS representations that allows for a controlled approximation error. We then show that this notion leads to a more efficient translation-invariant MPS representation of the W-state on a ring.

Let us first recall the definition of an MPS. To this end, we consider a state vector $T \in \left(\mathbb{C}^d\right)^{\otimes L}$ of $L$ spins of local dimension $d$. Expanding $T$ with respect to a product basis $\{|i_1, \ldots, i_l\rangle\}$ we obtain

$$T = \sum_{i_1, \ldots, i_L = 1}^{d} T_{i_1, \ldots, i_L} |i_1, \ldots, i_l\rangle, \tag{1}$$

with $T_{i_1, \ldots, i_L}$ denoting the basis coefficients. An MPS representation of $T$ can now be seen as a particular way of decomposing the order $L$ coefficient tensor $T_{i_1, \ldots, i_L}$ according to

$$T_{i_1, \ldots, i_L} = \text{tr}\left(M_{i_1}^{[1]} \cdots M_{i_L}^{[L]}\right), \tag{2}$$

with $M_{i_j}^{[j]}$ being a $D \times D$-matrix of sufficiently large dimension $D$, the so-called bond dimension. For each spin $j = 1, \ldots, L$, we can then define an order 3 tensor according to $M^{[j]} = \sum_{j=1}^{d} \sum_{\alpha, \beta = 1}^{D} (M_i^{[j]})_{\alpha, \beta} |\alpha\rangle\langle\beta| \otimes |i\rangle$ and by setting $A_j = \sum_{j=1}^{d} \sum_{\alpha, \beta = 1}^{D} (M_i^{[j]}) |i\rangle\langle\alpha\beta|$, we can in turn express the state vector $T$ as

$$T = \left(\bigotimes_{j=1}^{L} A_j\right)\left(\bigotimes_{k=1}^{L} \Omega_{k, k+1}^{D}\right), \quad \Omega^D = \sum_{l=1}^{D} |l, l\rangle, \tag{3}$$

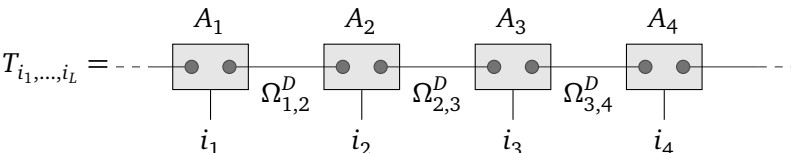

Figure 1: Matrix product states as a network of maximally entangled states $\Omega^D$ shared between physical sites of the 1D lattice to which local operations $A_j$ are applied on combined virtual space on each site.

which is an MPS representation with periodic boundary conditions and bond dimension $D$ of the state $T$. Note that the two tensor products in (3) are shifted with respect to each other by half a physical lattice site such that $\Omega_{k,k+1}$ corresponds to a maximally entangled state shared between the lattice sites $k$ and $k+1$, whereas $A_j$ acts on the combined virtual space $\mathbb{C}^D \otimes \mathbb{C}^D$ at lattice site $j$ (see also Figure 1). The important observation about (3) is the fact that we are applying these linear maps $A_j$ locally at each lattice site to an underlying resource state, which in the case of an MPS is given by maximally entangled pair states $\Omega^D$ shared between neighbouring lattice sites if we think of the $L$ spins positioned on a one-dimensional ring.

The question of whether a given vector $\psi$ living in a tensor product space $\bigotimes_{j=1}^{L} \mathbb{C}^{d_j}$ can be transformed into a state $\phi \in \bigotimes_{j=1}^{L} \mathbb{C}^{d'_j}$ via local linear maps $A_j : \mathbb{C}^{d_j} \mapsto \mathbb{C}^{d'_j}$ is known in the context of algebraic complexity theory as restriction.

**Definition 1** (Restriction). Given $\psi \in \bigotimes_{j=1}^{m} \mathbb{C}^{d_j}$ and $\phi \in \bigotimes_{i=j}^{m} \mathbb{C}^{d'_j}$ we say that $\psi$ restricts to $\phi$, denoted as $\psi \geq \phi$ if there exist linear maps $\{A_j : \mathbb{C}^{d_j} \mapsto \mathbb{C}^{d'_j}\}$ such that

$$\left( \bigotimes_{j=1}^{m} A_j \right) \psi = \phi \,. \tag{4}$$

Note that the domain of the local maps $A_j$ is implicitly specified via the chosen tensor decomposition $\bigotimes_{j=1}^{m} \mathbb{C}^{d_j}$ of the underlying Hilbert space, i.e. each $A_j$ acts on the corresponding tensor factor in this decomposition.

An important generalization of the concept of restriction is that of degeneration. Here, instead of an exact conversion according to (4), we allow for approximate conversions between a state $\psi$ to a state $\phi$ by local operations.

**Definition 2** (Degeneration). Let $\psi \in \bigotimes_{i=1}^{m} \mathbb{C}^{d_i}$ and $\phi \in \bigotimes_{i=1}^{m} \mathbb{C}^{d'_i}$ be pure states. We say that $\psi$ degenerates to $\phi$ with error degree $e$, denoted as $\psi \trianglerighteq^e \phi$, if there exist linear maps $A_i(\varepsilon) : \mathbb{C}^{d_i} \mapsto \mathbb{C}^{d'_i}$, depending polynomially on $\varepsilon$, such that

$$(A_1(\varepsilon) \otimes \cdots \otimes A_m(\varepsilon)) \psi = \varepsilon^d \phi + \sum_{l=1}^{e} \varepsilon^{d+l} \widetilde{\phi}_l, \tag{5}$$

for some tensors $\widetilde{\phi}_l$ and some integer $d$. We simply write $\psi \trianglerighteq \phi$ if $\psi \trianglerighteq^e \phi$ for some error degree $e$.

**Remark 3.** It is known (see e.g. [34]) that the definition of degeneration as given in (5) is equivalent to the following statement: $\psi \trianglerighteq \phi$ if there exists a sequence $\left( (A_j(n))_{j=1}^{m} \right)_n$ of linear maps $A_i(n) : \mathbb{C}^{d_i} \mapsto \mathbb{C}^{d'_i}$ such that

$$\lim_{n \to \infty} (A_1(n) \otimes \cdots \otimes A_m(n)) \psi = \phi \,.$$

**Remark 4.** The notion of degeneration is strictly weaker than that of restriction, in that given two vectors $\psi \in \bigotimes_{i=1}^m \mathbb{C}^{d_i}$, $\phi \in \bigotimes_{i=1}^m \mathbb{C}^{d_i'}$, $\psi$ can degenerate to $\phi$ even if we cannot find a restriction, i. e. $\psi \trianglerighteq \phi$, but $\psi \not\geq \phi$. A well known example of this fact is the degeneration from the GHZ-state on three qubits to the W-state [21, 35, 36] (see also the W-state example at the end of this section)

Let us denote by $\mathrm{GHZ}_k(m)$ the $k$-level Greenberger–Horne–Zeilinger (GHZ) state on $m$ parties:

$$\mathrm{GHZ}_k(m) = \sum_{i=1}^k \underbrace{|i\rangle \otimes \cdots \otimes |i\rangle}_{m \text{ times}}. \tag{6}$$

We note that $\mathrm{GHZ}_k$ agrees with the unit tensor in algebraic complexity theory, usually denoted as $\langle k \rangle$. In the cases when $m$ is small, as in the case $m = 3$ which we will study extensively, we will use the following graphical notation to represent the GHZ state:

$$\mathrm{GHZ}_k(3) = \sum_{i=1}^k |i\rangle |i\rangle |i\rangle = \boxed{k \triangleright} \cdot.$$

When the number of parties is clear from the context, we will simply write $\mathrm{GHZ}_k$ for simplicity.

Seen as the unit tensor, the GHZ state plays a special role in algebraic complexity theory, which leads us to define the following quantities.

**Definition 5** (Rank and border rank). For $\phi \in \bigotimes_{i=1}^m \mathbb{C}^{d_i}$ we define the *rank* and *border rank* of $\phi$ as

$$R(\phi) = \min\{k \in \mathbb{N}; \ \mathrm{GHZ}_k(m) \geq \phi\}, \tag{7}$$

$$\underline{R}(\phi) = \min\{k \in \mathbb{N}; \ \mathrm{GHZ}_k(m) \trianglerighteq \phi\}, \tag{8}$$

respectively.

**Remark 6.** Both the rank and the border rank depend on the tensor product structure of the space where $\phi$ lives: if we regroup the tensor product differently, the rank might change. It is easy to see that if we group factors together, i. e. we see $\phi$ not as an $m$-partite state but as an $m'$-partite state, with $m' < m$, then both the rank and the border rank will not increase. This is due to the fact that after regrouping the state $\mathrm{GHZ}_k(m)$ becomes the state $\mathrm{GHZ}_k(m')$, so if a restriction/degeneration to $\phi$ was possible before grouping it will still be possible after grouping.

Moreover, if $m = 2$, then both rank and border rank of $\phi$ coincide with the Schmidt rank across the bipartition. Therefore, we can see that the maximal Schmidt rank across any possible bipartition,

$$\mathrm{Sr}_{\max}(\phi) = \max_{K \subset \{1,\dots,m\}} \mathrm{rank}\, \mathrm{tr}_K\, |\phi\rangle\langle\phi| \,, \tag{9}$$

is a lower bound to $\underline{R}(\phi)$.

The question whether a given quantum state $\phi$ can be represented as an MPS with periodic boundary conditions of bond dimension $D$ is equivalent to the question, whether $\bigotimes_{k=1}^L \Omega_{k,k+1}^D \geq \phi$, where $\Omega_{k,k+1}^D$ again corresponds to a maximally entangled state with $D$ levels shared between the physical lattice sites $k$ and $k+1$ (see also Fig. 1). The state $\bigotimes_{k=1}^L \Omega_{k,k+1}^D$ is known in the context of algebraic complexity theory as the iterated matrix multiplication tensor, which is indeed the $L$-tensor given by maximally entangled states of dimensions $D_1$, $D_2$, …, $D_L$ arranged in a cycle. We will denote this tensor as $\mathrm{MaMu}_{D_1,\dots,D_L}$ (for Matrix Multiplication):

$$\mathrm{MaMu}_{D_1,\dots,D_L} = \sum_{i_1,\dots,i_L=1}^{D_1,\dots,D_L} |i_L i_1\rangle \otimes |i_1 i_2\rangle \otimes \cdots \otimes |i_{L-1} i_L\rangle. \tag{10}$$

This tensor is often denoted as $\langle D_1, \ldots, D_L \rangle$ in algebraic complexity. We write $\mathrm{MaMu}_D(L)$ if $D = D_1 = \cdots = D_L$, a case typically denoted as $\mathrm{IMM}_D^L$ in the literature. As in the case of the GHZ state, we will write $\mathrm{MaMu}_D$ without the parameter $L$ when this does not cause any ambiguity.

In the cases where $L$ is fixed and small, as for example when $L = 3$, we will use a similar graphical notation as for the GHZ-state:

$$\mathrm{MaMu}_{D_1,D_2,D_3} = \sum_{i_1,i_2,i_3=1}^{D_1,D_2,D_3} |i_1,i_2\rangle |i_2,i_3\rangle |i_3,i_1\rangle = \underset{D_3}{\overset{D_2}{D_1\rhd}} .$$

As mentioned before, $\mathrm{MaMu}_D(L)$ restricting to an $L$-tensor $\phi$ is equivalent to $\phi$ admitting an MPS representation of bond dimension $D$ with periodic boundary conditions. More generally, since PEPS and other tensor network states are defined in terms of networks of maximally entangled states, we will be interested in conversions between $\mathrm{MaMu}_D(L)$ and other states. This leads us to define, in analogy to the rank and border rank the following quantities.

**Definition 7** (Bond and border bond dimension)**.** For $\phi \in \bigotimes_{i=1}^m \mathbb{C}^{d_i}$ we define the (MPS) bond dimension and border bond dimension of $\phi$ as

$$\mathrm{bond}(\phi) = \min\{k \in \mathbb{N}; \ \mathrm{MaMu}_k(m) \geq \phi\}, \tag{11}$$

$$\underline{\mathrm{bond}}(\phi) = \min\{k \in \mathbb{N}; \ \mathrm{MaMu}_k(m) \unrhd \phi\}, \tag{12}$$

respectively.

**Remark 8.** Note that if we split the vertices $\{1, \ldots, m\}$ into $\{1, \ldots, r\}$ and $\{r+1, \ldots, m\}$ for some $r = 1, \ldots, m$, and we see $\mathrm{MaMu}_k(m)$ as a bipartite quantum state across this cut, the resulting state is equivalent to $\mathrm{MaMu}_k(2) = \mathrm{GHZ}_{k^2}(2)$ (since the MaMu tensor corresponds to periodic boundary conditions). Similarly to (9), we can consider the maximal Schmidt rank across any cut instead of any bipartition (i. e. we only consider bipartitions where the two parts are contiguous in the spin chain):

$$\mathrm{Sr}_{\mathrm{cut}}(\phi) = \max_{k \in \{1...m\}} \mathrm{rank} \, \mathrm{tr}_{1,...,k} \, |\phi\rangle\langle\phi| . \tag{13}$$

Then by the previous argument, we see that

$$\mathrm{Sr}_{\mathrm{cut}}(\phi)^{\frac{1}{2}} \leq \underline{\mathrm{bond}}(\phi) \leq \mathrm{bond}(\phi).$$

To conclude this section, we present an example where degenerations offer a more efficient state representation, i. e. an example where we have a separation between bond and border bond dimension if we require a translation invariant representation in both cases. To this end, consider the $W$-state on $L$ qubits defined as

$$W(L) = \sum_{\substack{i_1,\ldots,i_L=0 \\ i_1+\cdots+i_L=1}}^{1} |i_1, \ldots, i_L\rangle . \tag{14}$$

In the following, we give a translation-invariant representation of $W(L)$ with border bond dimension 2 independent of the system size $L$. This representation follows immediately from the well-known fact that the W-state (viewed as a homogenous polynomial $x^{L-1}y$) has border-Waring rank equal to two. For completeness we will give the argument below. In contrast to this border bond dimension 2 representation, the results from [37] on the quantum Wielandt inequality imply that the bond dimension of a translation-invariant restriction has to grow as

$\exp\left(\frac{1}{3}\omega(3L)\right)$, with $\omega(x)$ the product logarithm or Lambert function and it has been conjectured that the growth should be of the order $L^{1/3}$ [6]. We note however that without the restriction to the translation invariant setting one can also find a bond dimension 2 representation of the W-state.

In order to find a representation of the W-state $W(L)$ on $L$ parties with border bond dimension 2, note that

$$\psi_L(\varepsilon) = \begin{pmatrix} 1 & 0 \\ 0 & \varepsilon \end{pmatrix}^{\otimes L} |0\rangle^{\otimes L} = (|0\rangle + \varepsilon |1\rangle)^{\otimes L} = |0\rangle^{\otimes L} + \varepsilon W(L) + O(\varepsilon^2) \tag{15}$$

as a product state has bond dimension 1. Accordingly, the state $\widetilde{W}(\varepsilon, L) = \psi_L(\varepsilon) - |0\rangle^{\otimes L}$ is a degeneration from $\mathrm{MaMu}_L(2)$. The corresponding MPS-matrices of this translation-invariant border bond dimension 2 representation can be chosen as

$$M_0 = \begin{pmatrix} 1 & 0 \\ 0 & (-1)^{1/L} \end{pmatrix} \quad \text{and} \quad M_1 = \begin{pmatrix} \varepsilon & 0 \\ 0 & 0 \end{pmatrix}, \tag{16}$$

because $\mathrm{tr}\left(M_0^L\right) = 0$ and $\mathrm{tr}\left(M_0^n M_1^m\right) = \varepsilon^m$.

## 3 From PEPS to entanglement structures induced by hypergraphs

Going beyond one spatial dimension, the procedure described for MPS can be generalized to higher dimensional lattices which leads to the notion of projected entangled pair states (PEPS) [2]. Again maximally entangled states are shared with neighbouring lattice sites, and the local operations preparing the state of interest from this underlying resource state are allowed to operate on the combined virtual space that includes all these subsystems. This motivates the following definition of tensor networks and entanglement structures for general graphs.

**Definition 9** (Entanglement Structure (Graph)). Let $\mathfrak{G} = (V, E)$ be a graph with vertex set $V$, edge set $E$, and let $w$ be an integer-valued weight function on the edge set $w : E \to \mathbb{N}$. For each $e \in E$, let $\Omega_e \in \mathbb{C}^{w(e)} \otimes \mathbb{C}^{w(e)}$ be the maximally entangled state of Schmidt rank $w(e)$. An *entanglement structure* or *contraction scheme* w.r.t. to $\mathfrak{G}$ is then given by

$$\Psi(\mathfrak{G}) = \bigotimes_{e \in E} \Omega_e \in \bigotimes_{v \in V} \mathbb{C}^{D_v}, \tag{17}$$

with the local virtual dimension at vertex $v \in V$ given by $D_v = \prod_{e:v \in e} w(e)$. We call the *bond dimension* of $\Psi$ the quantity $\max_{v \in V}\{\sqrt[\deg(v)]{D_v}\}$.

For a fixed integer $D$ we will denote by $\Psi_D(\mathfrak{G})$ the entanglement structure obtained by setting a constant weight $w(e) = D$ on the graph (which will then have bond dimension $D$). We will also say that a state $\phi \in \bigotimes_{v \in V} \mathbb{C}^{d_v}$ is representable by $\mathfrak{G}$ with bond dimension $D$ iff $\Psi_D(\mathfrak{G}) \geq \phi$, where the locality structure of the restriction maps $A_j$ is given by vertex-set $V$ of $\mathfrak{G}$ according to the tensor decomposition $\bigotimes_{v \in V} \mathbb{C}^{D_v}$.

**Remark 10.** We remark that the term bond dimension is used in two different contexts. In Definition 9 it refers to the dimensionality of the maximally entangled states that form the graph entanglement structure, whereas in Definition 7 it characterizes for a given state vector the minimal bond dimension necessary to represent the state as an MPS. Note also that the notion of bond dimension and border bond dimension given in Definition 7 can be naturally extended to the case of entanglement structures defined on a general graph, i.e. as

$$\mathrm{bond}_{\mathfrak{G}}(\phi) = \min\{D \,|\, \Psi_D(\mathfrak{G}) \geq \psi\},$$

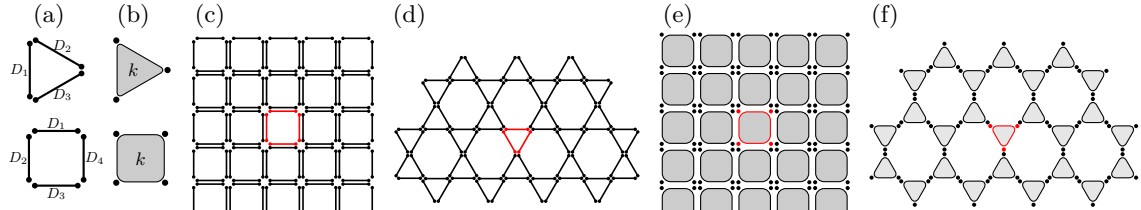

Figure 2: Examples of entanglement structures: (a) Plaquette tensors given by max-
imally entangled states shared cyclically between three and four sites, where the
indices $D_i$ denote the number of levels of the entangled states; (b) same as in (a) but
with a GHZ-state of $k$ levels shared between the sites; (c-d) entanglement structures
on the square and kagome lattice, the plaquette shown in red indicates how to obtain
those from the plaquette states from (a), neighbouring entangled states on the same
edge can be reinterpreted as a single maximally entangled state with the number of
levels squared; (e-f) same lattices as in (c-d) but with a generalized PEPS based on
3- and 4-party GHZ-states.

and similarly for $\underline{\mathrm{bond}}_{\mathfrak{G}}$. Since the tensor $\mathrm{MaMu}_k(m)$ can also be written as $\Psi_k(\mathfrak{C}_m)$, where $\mathfrak{C}_m$
is the cycle graph on $m$ vertices, Definition 7 of bond dimension and border bond dimension
coincide with $\mathrm{bond}_{\mathfrak{C}_m}$ and $\underline{\mathrm{bond}}_{\mathfrak{C}_m}$

Accordingly, MPS and PEPS fit naturally in this setting with the graph represented given
by the path graph $\mathfrak{L}_L$ in the case of open boundary MPS, by the cycle graph $\mathfrak{C}_L$ in the case of
periodic boundary MPS, and by a lattice graph in the case of PEPS, respectively (see Figure 2).
However, also more general tensor networks that allow for example for maximally entangled
states between next-to-nearest neighbours can be captured within this framework.

Definition 9 identifies the notion of representability again with the existence of a restriction
according to Definition 1, where the linear maps $\{A_i\}$ correspond to the local tensors defining
the tensor network state. We remark that our notion of bond dimension is chosen in such a
way that it captures how the number of parameters necessary to specify such a tensor network
state scales with the system size. More precisely, given the bond dimension $D$, the number of
parameters scales as $\mathcal{O}(|V|D^{\deg(\mathfrak{G})}d_{\max})$, where $d_{\max}$ is the maximal physical dimension given
by $\max_{i \in V}(d_i)$ and $\deg(\mathfrak{G})$ is the maximal degree of the vertices of $\mathfrak{G}$. This definition is general
enough to capture savings in the bond dimension due to non-uniform edge dimensions with
respect to the different edges in the graph, but at the same time reduces to the usual scaling
of $\mathcal{O}(|V|D^2 d_{\max})$ or $\mathcal{O}(|V|D^4 d_{\max})$ in the case of MPS or PEPS with uniform bond dimension,
respectively.

We will now generalize the concept of contraction schemes to representations of hyper-
graphs, where the underlying entanglement structure is given by multipartite entangled states
shared among all vertices that are connected by a hyperedge.

**Definition 11** (Entanglement Structure (Hypergraph)). Let $\mathfrak{G} = (V, E)$ be a hypergraph, with
vertex-set $V$ and hyperedge set $E$. For each $e \in E$, let $\Omega_e \in \bigotimes_{v \in e} \mathbb{C}^{D_{v,e}}$ be a pure state. An
*entanglement structure* or *contraction scheme* w.r.t. to $\mathfrak{G}$ is then given by

$$\Psi(\mathfrak{G}) = \bigotimes_{e \in E} \Omega_e \in \bigotimes_{v \in V} \mathbb{C}^{D_v},$$

with the local virtual dimension at vertex $v \in V$ given by $D_v = \prod_{e : v \in e} D_{v,e}$ and the bond
dimension of $\Psi(\mathfrak{G})$ defined as $D = \max_{v \in V} \{\sqrt[\deg(v)]{D}\}$.

Note that, contrary to the graph case, the hypergraph entanglement structure is not simply
defined by weights on the hyperedges but also by the choice of multi-partite entangled states

$\Omega_e$ (since there exist non-equivalent multi-partite entangled states, we cannot simply specify the edge dimension as in the case of graphs). As an example, note that the $\text{GHZ}_k(m)$ can be written as an entanglement structure on the hypergraph $\mathfrak{H}_m$ with $m$ vertices and a single hyperedge containing all vertices:

$$V(\mathfrak{H}_m) = \{0, \dots, m-1\}, \quad E(\mathfrak{H}_m) = \{V\},$$

by choosing $\Omega_V = \text{GHZ}_k(m)$.

In analogy to the graph case, we can still consider a hypergraph entanglement structure $\Psi(\mathfrak{G})$ as a contraction scheme, with a state $\phi$ being representable by $\Psi(\mathfrak{G})$ iff we can find local maps $\{A_v : \mathbb{C}^{D_v} \mapsto d_v\}$ satisfying (4) (i. e. iff $\Psi(\mathfrak{G}) \geq \phi$). Note that as in the graph case, the locality structure of the restriction maps $A_j$ is given by vertex-set $V$ of $\mathfrak{G}$ according to the tensor decomposition $\bigotimes_{v \in V} \mathbb{C}^{D_v}$.

Particular examples of entanglement structures on hypergraphs from the literature are projected entangled simplex states [28] and semi-injective PEPS [27]. In the latter case, the vertex set is given by the same vertices of the two-dimensional square lattice on $L \times L$ sites (i. e. $\mathfrak{C}_L \times \mathfrak{C}_L$), but instead of having an edge for each pair of neighbouring sites, there is instead a hyperedge containing the 4 vertices in each of the plaquettes:

$$V = [0, L] \times [0, L] \cap \mathbb{Z}^2,$$
$$e \in E \iff e = \{(i, j), (i+1, j), (i, j+1), (i+1, j+1)\} \quad \text{for some} \quad (i, j) \in V.$$

Finally, for each hyperedge $e$, we choose a GHZ state on 4 parties as $\Omega_e$, so that the resulting entanglement structure is given by

$$\Phi = \bigotimes_{e \in E} \text{GHZ}_k(4).$$

The bond dimension of $\Phi$ is then simply given by the number of GHZ levels $k$. For $k = 2$ this class of states also includes the ground state of the CZX-model, exhibiting an on-site $\mathbb{Z}_2$ symmetry [26].

In order to find representations of physical states with optimal bond dimension, we will analyze how well a given contraction scheme can be expressed in terms of another. To this end, we introduce the following definition, which specializes Definition 1 to the particular case of entanglement structures.

**Definition 12** (Conversion of entanglement structures)**.** Let $\mathfrak{G}$ and $\mathfrak{G}'$ be two graphs or hypergraphs with the same vertex set $V$. Given two entanglement structures $\Psi(\mathfrak{G})$ and $\Psi(\mathfrak{G}')$ we say that $\Psi(\mathfrak{G})$ restricts to $\Psi(\mathfrak{G}')$, and we write $\Psi(\mathfrak{G}) \geq \Psi(\mathfrak{G}')$, if there exist linear maps $A_v : \mathbb{C}^{D_v} \to \mathbb{C}^{D_v'}$ for each $v \in V$ such that

$$\left( \bigotimes_{v \in V} A_v \right) \Psi(\mathfrak{G}) = \Psi(\mathfrak{G}'), \tag{18}$$

where $D_v$ and $D_v'$ are the local dimension at vertex $v$ of $\Psi(\mathfrak{G})$ and $\Psi(\mathfrak{G}')$, respectively. The notion of degeneration specializes to the case of entanglement structures exactly in the same way as restrictions (i. e. by allowing local maps to act according to the tensor product structure defined by the vertex set $V$).

**Remark 13.** Note that in the case of a path graph $\mathfrak{L}_L$ on $L$ sites and a graph entanglement structure $\Psi_k(\mathfrak{L}_L)$ (i. e. the entanglement structure of an open boundary condition MPS of bond dimension $k$), the existence of a degeneration implies the existence of a restriction. More concretely, if $\Psi_k(\mathfrak{L}_L) \unrhd T$ for some $L$-partite quantum state $T$, then also $\Psi_k(\mathfrak{L}_L) \geq T$. This

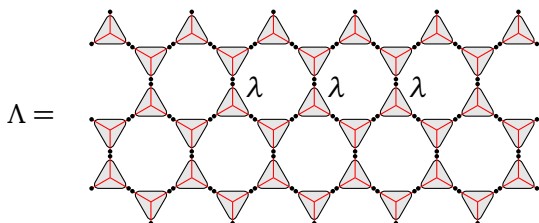

Figure 3: The hypergraph entanglement structure $\Lambda$ of the RVB state. The triangles $\triangleright$ represent the $\lambda$ tensor, and the entanglement structure $\Lambda$ is obtained by tensoring $\frac{2}{3}L$ copies of it and arranging the vertices according to the kagome lattice.

is due to the fact that, by sequential SVD decompositions (see [6, Theorem 1] and [4, pag. 18-20]), it is possible to construct $T$ with a bond dimension equal to the maximal Schmidt rank across any cut $\mathrm{Sr}_{\mathrm{cut}}(T)$ (see (13)): equivalently $\Psi_k(\mathfrak{L}_L) \geq T$ for $k = \mathrm{Sr}_{\mathrm{cut}}(T)$. On the other hand we can repeat the argument of Remark 8 for $\Psi_k(\mathfrak{L}_L)$, but taking into account that we have open boundary conditions instead: we see that after grouping neighbouring sites we can convert $\Psi_k(\mathfrak{L}_L)$ to $\Psi_k(\mathfrak{L}_{L'})$ with $L' < L$, so that if $\Psi_k(\mathfrak{L}_L) \unrhd T$ then necessarily $k \geq \mathrm{Sr}_{\mathrm{cut}}(T)$. On the other hand, as soon as there are cycles in the graph, this argument breaks down, and we have already seen in the W-state example at the end of Section 2 that a degeneration can exist when the corresponding restriction does not.

**RVB state**    Another example of entanglement structure is found in the context of PEPS representations of the Resonating Valence Bond state (RVB) [31–33]. Rephrasing the construction used in [32] in terms of Definition 11, the RVB state on the kagome lattice can be represented by an entanglement structure $\Lambda$, where we assign to each plaquette the 3-party entangled state $\lambda \in (\mathbb{C}^3)^{\otimes 3}$ (see Figure 3) given by

$$\lambda = \sum_{i,j,k=0}^{2} \epsilon_{i,j,k} |i,j,k\rangle + |2,2,2\rangle = \triangleright, \tag{19}$$

where $\epsilon_{i,j,k}$ denotes the antisymmetric tensor with $\epsilon_{0,1,2} = 1$.

In [32] the entanglement structure $\Lambda$ for the RVB state composed of plaquette tensors $\lambda$ was shown to have a PEPS representation with bond dimension 3, by constructing the explicit linear maps realizing the conversion. The RVB state on the kagome lattice thus has bond dimension at most 3. We now give a representation of the same state with border bond dimension equal to 2 (in other words we reduce the local virtual dimension at each vertex from $3^4 = 81$ to $2^4 = 16$), which, as we show in Section 5.3, is smaller than the optimal PEPS representation that can be obtained with restrictions.

In order to do so, we need to show that $_2\triangleright_2^2 \unrhd \triangleright$. The MPS matrices of the degeneration are given by

$$M_0^{[j]}(\epsilon) = \frac{1}{2}\begin{pmatrix} 0 & \epsilon \\ \epsilon & 0 \end{pmatrix}, \quad M_1^{[j]}(\epsilon) = \begin{pmatrix} 0 & -\epsilon \\ \epsilon & 0 \end{pmatrix}, \quad M_2^{[j]}(\epsilon) = \begin{pmatrix} 1 & 0 \\ 0 & -1 \end{pmatrix} + \frac{\delta_{j,3}\epsilon^2}{2}\begin{pmatrix} 1 & 0 \\ 0 & 1 \end{pmatrix}, \tag{20}$$

with $j = 1, 2, 3$, resulting in the state $\varepsilon^2 \lambda + \varepsilon^4 |2\rangle \otimes (\frac{1}{4}|00\rangle - |11\rangle)$.

In the following section, we will show how this improved representation can be used to compute exact contractions and expectation values for the RVB state.

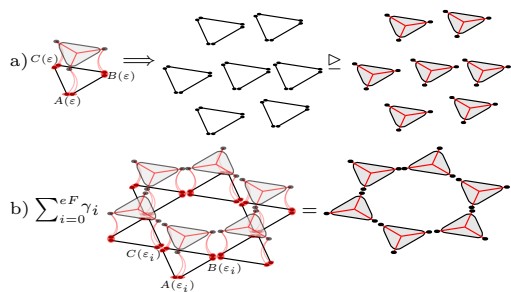

Figure 4: Graphical representation of Theorem 14: a) A local degeneration $(A(\varepsilon), B(\varepsilon), C(\varepsilon))$ depending polynomially on $\varepsilon$ from one plaquette state (pairwise entangled states between three parties) to another ($\lambda$ state), gives rise to a global degeneration between a collection of $F$ plaquette states. b) Evaluating the degeneration at $eF + 1$ points $\varepsilon_i$, we can express the full entanglement structure built from the second plaquette state (here $\lambda$ states) as a superposition of $eF + 1$ states that arise as restrictions from the first entanglement structure (here pairwise entangled states between three parties). The parameter $e$ is a scaling factor depending on the polynomial degree of the local degeneration, the prefactor $\gamma_i$ is obtained by evaluating the $i$th Lagrange polynomial $\ell_i$ at 0.

## 4 Exact state representations from degenerations

Having introduced the concept of tensor network representations in terms of degenerations and border bond dimension in the previous sections, we will now turn to the question how to obtain physical information in this approximate setting. To this end, we present a general method to turn an approximate conversion between two entanglement structures given by degenerations into an exact one by allowing for superpositions. The main advantage of this approach is on the one hand that this can be accomplished with only a linear overhead in the number of plaquette states involved and on the other hand that it also allows for the computation of exact expectation values. These properties are summarized in the following theorem. The proof relies on results from algebraic complexity theory [38,39] and we include the argument here for the sake of completeness.

**Theorem 14.** *Let $\Psi$ and $\Phi$ be the entanglement structures obtained by placing $\psi$ and $\phi$, respectively, on $F$ faces of a lattice with $L$ sites. Assume $T$ can be represented by $\Phi$, i. e. $\Phi \geq T$. If $\psi \trianglerighteq \phi$, then*

$$T = \sum_{i=0}^{eF} W_i, \tag{21}$$

*where each $W_i$ can be represented by $\Psi$, i. e. $\Psi \geq W_i$. The number of terms in the representation is linear in $F$, i. e. the constant $e$ is only dependent on the degeneration $\psi \trianglerighteq \phi$. Moreover expectation values of an observable $O$ under $T$ can be computed from expectation values of $2eF + 1$ states represented by $\Psi$:*

$$\langle T, OT \rangle = \sum_{i=0}^{2eF} \gamma_i \langle V_i, OV_i \rangle, \tag{22}$$

*where again each $V_i$ can be represented by $\Psi$, i.e. $\Psi \geq V_i \ \forall i$, and $\gamma_i \in \mathbb{R}$ are known constants depending on $V_i$.*

*Proof.* According to Definition 2, $\psi \rhd \phi$ if there exist linear maps $A_i(\epsilon) : \mathbb{C}^{d_i} \mapsto \mathbb{C}^{d'_i}$, depending polynomially on the parameter $\epsilon$, such that

$$(A_1(\epsilon) \otimes \cdots \otimes A_m(\epsilon))\psi = \varepsilon^d \phi + \sum_{l=1}^{e} \epsilon^{d+l} \widetilde{\phi}_l,$$

for some multi-partite states $\widetilde{\phi}_l$, approximation degree $d$ and error degree $e$. We observe that the plaquette degeneration $\phi \rhd \psi$ immediately implies that $\phi^{\otimes F} \rhd \psi^{\otimes F}$, as can be seen by taking the tensor product of the local operators given by the degeneration $\phi \rhd \psi$. As was already observed in [39, Prop. 4], the error degree will only grow linearly in the number of copies of the degeneration maps, i.e. the number of faces $F$ in the lattice and therefore we see that the product of $F$ copies of $\phi$ degenerates to $F$ copies of $\psi$ with error degree $eF$:

$$\underbrace{\psi \otimes \psi \otimes \cdots \otimes \psi}_{F \text{ copies}} \rhd^{eF} \varphi \otimes \varphi \otimes \cdots \otimes \varphi. \tag{23}$$

This degeneration is possible, when all $m$ parties of each of the $F$ copies are considered independently, i.e. when the states in (23) are regarded are $mF$-partite states. In [39] this was derived in order to show that tensor rank is strictly submultiplicative under the tensor product. Note that the degeneration resulting from grouping all the $F$ copies of $\psi$ and $\phi$ into an $m$-tensor was already considered in [38], and led to faster algorithms for matrix multiplication. In order to prove the theorem, we will consider instead a different consequence of this argument: grouping the $mF$ tensor factors according to the underlying lattice, we obtain $\Psi$ and $\Phi$ as $L$-partite states respectively, which means that

$$\psi \rhd^e \phi \implies \Psi \rhd^{eF} \Phi. \tag{24}$$

Similar as in [38, 39], we now apply Lagrange interpolation [40, p. 260] in order to transform the degeneration into a restriction. From (24), we can write

$$\left( \bigotimes_{i=1}^{F} \left( \bigotimes_{l=1}^{m} A_l(\varepsilon) \right) \right) \Psi = \varepsilon^d \Phi + \sum_{k=1}^{eF} \varepsilon^{d+k} \widetilde{\Phi}_k \tag{25}$$

for some integer $d$, where the linear maps $A_l(\varepsilon)$, depending polynomially on $\varepsilon$, are given by copies of the degeneration maps of $\psi \rhd \phi$ corresponding to the plaquettes $f = 1, \ldots, F$. Let $(B_l)_l$ be the local operators given by the restriction $\Phi \geq T$ at the lattice sites $l = 1, \ldots L$, i.e. $\left( \bigotimes_{i=1}^{L} B_l \right) \Phi = T$. Composing (24) with the $(B_l)_l$ and dividing by $\epsilon^d$, we define

$$T(\epsilon) = \epsilon^{-d} \left( \bigotimes_{i=1}^{L} B_l \right) \left( \bigotimes_{i=1}^{F} \left( \bigotimes_{l=1}^{m} A_l(\varepsilon) \right) \right) \Psi = T + \sum_{k=1}^{eF} \varepsilon^k \widetilde{T}_k. \tag{26}$$

Considering the right hand side, we immediately see that $T(\epsilon)$ depends polynomially on $\epsilon$ with degree $eF$, and that $T(0) = T$. Moreover, for each $\epsilon \neq 0$, $T(\epsilon)$ is a restriction from $\Psi$. Evaluating $T(\epsilon)$ at $eF + 1$ points $(\epsilon_i)_{i=0}^{eF}$, we can obtain the value at $\epsilon = 0$ via Lagrange interpolation:

$$T = T(0) = \sum_{i=0}^{eF} \gamma_i T(\epsilon_i),$$

where $\gamma_i = \ell_i(0)$ is obtained by evaluating the $i$th Lagrange polynomial $\ell_i$ at 0. Defining $W_i = \gamma_i T(\epsilon_i)$, we obtain (21). In order to prove (22), we observe that any expectation value with respect to $T(\epsilon)$ is given by

$$\langle T(\epsilon), OT(\epsilon) \rangle = \langle T, OT \rangle + \sum_{k=1}^{eF} \left( \langle T, O\tilde{T}_k \rangle \varepsilon^k + \langle \tilde{T}_k, OT \rangle (\overline{\epsilon})^k \right) + \sum_{k,k'=1}^{eF} \langle \tilde{T}_{k'}, O\tilde{T}_k \rangle (\overline{\epsilon})^k \varepsilon^{k'}. \tag{27}$$

In case $\varepsilon \in \mathbb{R}$ this is again a polynomial in $\varepsilon$ now of degree $2eF$. Similarly as before, computing $\langle T(\epsilon), OT(\epsilon) \rangle$ for a fixed $\epsilon \in \mathbb{R}$ amounts to computing an expectation value for a state $T(\epsilon)$ which has a representation in terms of $\Psi$. Computing $2eF + 1$ of such expectation values is then again sufficient for computing $\langle T(\epsilon), OT(\epsilon) \rangle$ at $\epsilon = 0$ via interpolation, this proves (22). $\qquad\square$

Let us note that we are not limited to use $\varepsilon \in \mathbb{R}$, but can also choose $\varepsilon \in \mathbb{C}$ if that is more convenient. To this end, let us consider the expression

$$\langle T(\overline{\varepsilon}), OT(\varepsilon) \rangle = \langle T, OT \rangle + \sum_{k=1}^{eF} \left( \langle T, O\tilde{T}_k \rangle + \langle \tilde{T}_k, OT \rangle \right) \varepsilon^k + \sum_{k,k'=1}^{eF} \langle \tilde{T}_{k'}, O\tilde{T}_k \rangle \varepsilon^{k'+k}.$$

This is by design again a $2F$ degree polynomial in $\varepsilon$ with the expectation value $\langle T, OT \rangle$ in leading order. Hence, computing the scalar product $\langle T(\overline{\varepsilon}_i), OT(\varepsilon_i) \rangle$ for $2F + 1$ different values of $\varepsilon_i$ will again allow us to compute the value of this polynomial at $\varepsilon = 0$ and therefore $\langle T, OT \rangle$. Alternatively, we could also just insert $\varepsilon$ directly into (27) and treat $\text{Re}(\varepsilon)$ and $\text{Im}(\varepsilon)$ as independent variables.

We also note that due to the reduced bond dimension for each of the $V_i$ also the error caused by approximate contraction will be smaller and that by oversampling the number of evaluation points in the degeneration, there is an additional potential for improving the accuracy of the contraction.

Theorem 14 provides degenerations that transform the two entanglement structures into each other plaquette by plaquette. Constructions based on larger units (e.g. several plaquettes) might lead to further reductions in the bond dimension, since the maps on the vertices that are grouped together no longer have a tensor product constraint.

Before looking more generally on plaquette conversions between important classes of tensor networks in the next section, as a first application of the theorem we come back to the RVB state on the kagome lattice in terms of the $\Lambda$ entanglement structure introduced at the end of section 3. We have presented a degeneration from $_2\!\!\!\Large\triangleright_2$ to the plaquette tensors $\triangleright$ of $\Lambda$, which has approximation degree $d$ and error degree $e$ both equal to 2. Rolling this out on the kagome lattice with $F$ triangles, we obtain a border PEPS representation of the RVB state of border bond dimension 2 and Theorem 14 then ensures that we can reconstruct the RVB state as a superposition of $2F + 1$ PEPS of bond dimension 2 or compute expectation values with $4F + 1$ contractions.

# 5 Plaquette conversions

In this section, we present general strategies and examples for optimized conversion between plaquette states in terms of degenerations. To this end, we consider $m$-tensors, i.e. elements of $\bigotimes_{i=1}^{m} \mathbb{C}^{d_i}$, for some non-zero integers $(d_i)_i$, which can be equivalently seen as unnormalized pure $m$-partite quantum states. We will usually consider $m$ to be a small integer (often $m$ will be equal to 3 or 4), as these $m$-tensors are the building blocks of the entanglement structures we considered in Section 3. After some definitions and examples that set the scene, we will study the conversion between maximally entangled states shared around circles and GHZ states, which are the basis for conversion between PEPS and more general tensor network states. To do this, we utilize the correspondence between entangled pairs on the circle and the matrix multiplication tensor (see e.g. [41]). This will be first done for 3-party tensors and subsequently for tensors of $m$ parties. In addition, we prove in Section 5.3 that the MPS representation with bond dimension $(2, 2, 3)$ for the state $\lambda$, which is the basis for the PEPS representation of the RVB state, is optimal.

## 5.1 From $\mathrm{MaMu}_k(m)$ to $\mathrm{GHZ}_k(m)$: the case $m = 3$

The aim of this section is to investigate restrictions and degenerations from $\mathrm{MaMu}_{k_1,k_2,k_3}$ to $\mathrm{GHZ}_k(3)$ and viceversa: this will allow us to express GHZ based hypergraph entanglement structures on triangular lattices as bond and border bond PEPS representations. In particular, we will prove the following proposition.

**Proposition 15.**

$$\frac{1}{2}\Big(1 + \sqrt{4k-3}\Big) < \mathrm{bond}(\mathrm{GHZ}_k(3)) \leq \mathcal{O}\Big(k^{\frac{1}{2}+\frac{c}{\sqrt{\log k}}}\Big), \tag{28}$$

*for some fixed positive c. In other words,*

$$\mathrm{MaMu}_n(3) \not\geq \mathrm{GHZ}_{n^2-n+1}(3) \quad and \quad \mathrm{MaMu}_n(3) \geq \mathrm{GHZ}_{f(n)}(3), \tag{29}$$

*where $f(n) = \mathcal{O}\left((n^2)^{1-\frac{c'}{\sqrt{\log n}}}\right)$ for some positive constant $c'$.*

However, as it was shown by Strassen in [42, Thm. 6.6], there exist degenerations which allow for an MPS representation of $\mathrm{GHZ}_{\lceil \frac{3}{4}n^2\rceil}(3)$ with border bond dimension $n$. Setting $n = 2$, this shows in particular, that

$$2 \rhd\!\!\!\!\!\!\!{}^{2}_{2} \not\geq \boxed{3}\!\!\rhd\cdot \quad \text{but} \quad 2 \rhd\!\!\!\!\!\!\!{}^{2}_{2} \trianglerighteq \boxed{3}\!\!\rhd\cdot .$$

Hence, $\mathrm{bond}(\mathrm{GHZ}_3(3)) > 2$, whereas $\underline{\mathrm{bond}}(\mathrm{GHZ}_3(3)) = 2$.

Before giving the proof, we discuss a non-symmetric extension of this result, i.e. degenerations from $\mathrm{MaMu}_{k_1,k_2,k_3}$ with different values of $k_1$, $k_2$, $k_3$. Following [43], we consider the local diagonal operator

$$A(\epsilon)|i,j\rangle = \epsilon^{(i-g)^2+2ij}|i,j\rangle \tag{30}$$

depending on an integer $g$ which we will fix later. This leads to the transformation

$$(A(\varepsilon) \otimes A(\varepsilon) \otimes A(\varepsilon))\,\mathrm{MaMu}_{k_1,k_2,k_3} = \epsilon^{2g^2}\sum_{i_1,i_2,i_3=1}^{k_1,k_2,k_3} \epsilon^{(i_1+i_2+i_3-g)^2}|i_1,i_2,\rangle|i_2,i_3\rangle|i_3,i_1\rangle$$

$$= \epsilon^{2g^2}\sum_{\substack{i_1,i_2,i_3=1\\i_1+i_2+i_3=g}}^{k_1,k_2,k_3} |i_1,i_2\rangle|i_2,i_3\rangle|i_3,i_1\rangle + \mathcal{O}\Big(\epsilon^{2g^2+1}\Big).$$

The leading order term in $\epsilon$ corresponds to a GHZ state, because fixing any pair of $i_1$, $i_2$, $i_3$ determines the third one uniquely. Hence, we only have to determine the number of solutions to the equation $i_1 + i_2 + i_3 = g$ for given $n_i$ and inhomogeneity $g$. Choosing $k_1 = 2$, $k_3 = 3$ and $k_2 = 2$ or $k_2 = 3$ and $g = 5$ then directly leads to

$$2 \rhd\!\!\!\!\!\!\!{}^{2}_{3} \trianglerighteq \boxed{4}\!\!\rhd\cdot \quad \text{and} \quad 2 \rhd\!\!\!\!\!\!\!{}^{3}_{3} \trianglerighteq \boxed{5}\!\!\rhd\cdot .$$

These degenerations are optimal, both in the sense that the corresponding restrictions are not possible, and in the sense that we cannot obtain GHZ states with more levels from a degeneration of these MaMu tensors. It is also not possible to obtain the same GHZ states from MaMu tensors, where one of the bond dimension is smaller than the ones we have considered.

We will now turn to the proof of Proposition 15. We will first introduce two definitions and prove a lemma.

**Definition 16.** Let $\mathfrak{G} = (V, E)$ be a graph. An *orthogonal representation* of $\mathfrak{G}$ is a mapping

$$\pi : V \to \mathcal{H} \setminus \{0\},$$

from the graph into some inner product vector space $\mathcal{H}$ such that

$$(u, v) \in E \implies \langle \pi(u) | \pi(v) \rangle_{\mathcal{H}} = 0.$$

We will denote by $\dim \mathcal{H}$ the dimension of the orthogonal representation.

**Definition 17.** Let $\mathfrak{K}_{n,n}$ be the complete bipartite graph on $2n$ vertices, i. e.

$$V(\mathfrak{K}_{n,n}) = \{b_0, \ldots, b_{n-1}, c_0, \ldots, c_{n-1}\},$$
$$E(\mathfrak{K}_{n,n}) = \{(b_j, c_k) \mid j, k = 0, \ldots, n-1\}.$$

Let $\mathfrak{K}_{n,n}^0$ be the graph obtained by removing the edge $(b_0, c_0)$ from $\mathfrak{K}_{n,n}$:

$$E(\mathfrak{K}_{n,n}^0) = E(\mathfrak{K}_{n,n}) \setminus \{(b_0, c_0)\} = \{(b_j, c_k) \mid j, k = 0, \ldots, n-1, j \neq k \text{ or } j = k \neq 0\}.$$

**Lemma 18.** *With the notation defined above, let $\pi : \mathfrak{K}_{n,n}^0 \to \mathcal{H}$ be an orthogonal representation such that $\dim \mathcal{H} \leq 2(n-1)$. Then at least one of the following holds*

1. $\dim \operatorname{span}\{\pi(b_i) \mid i = 1, \ldots, n-1\} < n-1$,

2. $\dim \operatorname{span}\{\pi(c_i) \mid i = 1, \ldots, n-1\} < n-1$,

3. $\pi(b_0)$ *is orthogonal to* $\pi(c_0)$.

*Proof.* Let $\mathcal{B} = \operatorname{span}\{\pi(b_i) \mid i = 1, \ldots, n-1\}$ and $\mathcal{C} = \operatorname{span}\{\pi(c_i) \mid i = 1, \ldots, n-1\}$. Since $\pi(b_i)$ is orthogonal to $\pi(c_j)$ for every $i, j = 1, \ldots, n-1$, we have that $\mathcal{B} \perp \mathcal{C}$. If $\mathcal{B} \oplus \mathcal{C}$ is not equal to $\mathcal{H}$, which has dimension $\leq 2(n-1)$, then at least one of the two has to have dimension strictly smaller than $n-1$, so that either *1.* or *2.* holds. If not, then $\mathcal{H} = \mathcal{B} \oplus \mathcal{C}$. Since $\pi(b_0)$ is orthogonal to every $\pi(c_i)$ for $i = 1, \ldots, n-1$, it is orthogonal to $\mathcal{C}$, and therefore $\pi(b_0) \in \mathcal{B}$. Similarly, $\pi(c_0)$ is orthogonal to $\mathcal{B}$ and therefore lies in $\mathcal{C}$. But then $\pi(b_0)$ and $\pi(c_0)$ live in orthogonal subspaces and they are themselves orthogonal. $\qquad\square$

We are now ready to prove Proposition 15.

*Proof (Proposition 15).* We will start by proving the lower bound of (28) as well as first part of (29), since they are equivalent as can be seen by setting $k = n^2 - n + 1$. Let us assume that $\text{GHZ}_{n^2-n+1}$ has an MPS representation with bond dimension $D \leq n$, and let us show how to derive a contradiction from this fact. To fix notation, let

$$\text{GHZ}_{n^2-n+1} = \sum_{i,j,k=0}^{n^2-n} \operatorname{tr}\left[A_i B_j C_k\right] |i, j, k\rangle,$$

for some non-zero matrices $\{A_i\}_i, \{B_j\}_j$ and $\{C_k\}_k$ of dimension $D \times D$, such that

$$\operatorname{tr} A_i B_j C_k = \begin{cases} 1 & \text{if} \quad i = j = k, \\ 0 & \text{otherwise.} \end{cases}$$

We start by showing that if $D \leq n$ we can without loss of generality assume that $A_0$ is non-singular. To derive this, we will use the following fact: any linear subspace of $\mathcal{M}_D$ containing only singular matrices has dimension at most $D^2 - D$ [44]. Consider $S = \operatorname{span}\{A_i \mid i = 0, \ldots, n^2 - n\} \subset \mathcal{M}_D$. $S$ is the span of $n^2 - n + 1$ matrices: if it contains

only singular matrices, then its dimension can be at most $D^2 - D$. So if $D \leq n$, either in $S$ there is one matrix which has full rank or $\dim S \leq D^2 - D \leq n^2 - n$, which implies that the matrices $(A_i)_i$ are not linearly independent.

Let $W = (w_{ij}) \in U(n^2 - n + 1)$ a unitary matrix such that $\sum_{i=0}^{n^2-n} w_{0i} A_i$ is either zero or full rank. Then by denoting $\phi_i = W |i\rangle$ the rotated basis, we see that $(W \otimes W \otimes W) \mathrm{GHZ}_{n^2-n+1} = \sum_i \phi_i \otimes \phi_i \otimes \phi_i$ has an MPS representation with matrices

$$\tilde{A}_i = \sum_j w_{i,j} A_j, \quad \tilde{B}_i = \sum_j w_{i,j} B_j, \quad \tilde{C}_i = \sum_j w_{i,j} C_j,$$

and $\tilde{A}_0$ is either zero or full-rank. The first case we can exclude, because $\operatorname{tr} \tilde{A}_0 \tilde{B}_0 \tilde{C}_0 = 1$. This shows that up to a local unitary on the physical level, we can assume without loss of generality that $A_0$ is not singular.

Let $A_0 = U \Sigma V^*$ be the singular-value decomposition of $A_0$. Then $\Sigma > 0$ defines a scalar product on $\mathcal{M}_D \simeq \mathbb{C}^{D^2}$ by $\langle X | Y \rangle_\Sigma = \operatorname{tr} \Sigma X^* Y$. Defining

$$\pi(b_j) = B_j^* V, \quad \pi(c_k) = C_k U, \quad j, k = 0 \ldots n^2 - n,$$

we obtain an orthogonal representation of the graph $\mathfrak{K}_{n^2-n+1, n^2-n+1}^0$ (defined in Lemma 18) on $\mathcal{M}_D$ with inner product $\langle \cdot | \cdot \rangle_\Sigma$, since

$$\langle \pi(b_j) | \pi(c_k) \rangle_\Sigma = \operatorname{tr} \Sigma V^* B_j C_k U = \operatorname{tr} A_0 B_j C_k = \begin{cases} 1 & \text{if } j = k = 0, \\ 0 & \text{otherwise.} \end{cases}$$

If $D \leq n$, then $\dim \mathcal{M}_D = D^2 \leq n^2 < n^2 + (n-1)^2 + 1 = 2(n^2 - n + 1)$, which implies that we can apply Lemma 18 and at least one of the conditions stated in it must hold true. If $1.$ or $2.$ hold, then either $\operatorname{span}\{B_i\}$ or $\operatorname{span}\{C_i\}$ has dimension strictly smaller than $n^2 - n + 1$, but we have already seen that this leads to a contradiction. Therefore $3.$ must hold, but this also leads to a contradiction: on the one hand we have proven that $\operatorname{tr} A_0 B_0 C_0 = 0$ but we also know that know that $\operatorname{tr} A_0 B_0 C_0 = \langle \pi(b_0) | \pi(c_0) \rangle = 1$.

We will now prove the upper bound of (28). Our starting point is the following result [42, Thm. 6.6]

$$\mathrm{MaMu}_n \trianglerighteq^{\gamma n^2} \mathrm{GHZ}_{\lceil 3n^2/4 \rceil}, \tag{31}$$

for some constant $\gamma > 0$. Let $\alpha$ an integer to be determined later, and consider the tensor product of $\alpha$ copies of (31). To simplify notation, we set $k = (\lceil \frac{3}{4} n^2 \rceil)^\alpha$, so that we get

$$\mathrm{MaMu}_{n^\alpha} \trianglerighteq^{\alpha \gamma n^2} \mathrm{GHZ}_k.$$

As we have discussed previously, it is a well known result in algebraic complexity theory that a degeneration can be turned into a restriction by interpolation paying a price in terms of a direct sum (see e.g. [34]). In the present context, this means that we can turn the degeneration into a restriction by supplementing a GHZ state with a number of levels equal to the error degree plus one (see e.g. [39]). Therefore we obtain

$$\mathrm{GHZ}_{\alpha \gamma n^2 + 1} \otimes \mathrm{MaMu}_{n^\alpha} \geq \mathrm{GHZ}_k,$$

from which follows that

$$\mathrm{bond}(\mathrm{GHZ}_k) \leq n^\alpha \, \mathrm{bond}(\mathrm{GHZ}_{\alpha \gamma n^2 + 1}).$$

We can trivially bound $\mathrm{bond}(\mathrm{GHZ}_{\alpha \gamma n^2 + 1})$ by $2\alpha \gamma n^2$,

$$\mathrm{bond}(\mathrm{GHZ}_k) \leq 2\alpha \gamma n^{\alpha + 2}. \tag{32}$$

From the definition of k, we have $n \leq \left(\frac{4}{3}\right)^{\frac{1}{2}} k^{\frac{1}{2\alpha}}$, and by inserting this into the right-hand side of (32), we obtain:

$$\text{bond}(\text{GHZ}_k) \leq 2\gamma\alpha\left(\frac{4}{3}\right)^{1+\frac{\alpha}{2}} k^{\frac{1}{2}+\frac{1}{\alpha}}. \tag{33}$$

We now want to choose $\alpha$ in order to minimize the right hand side. We will instead simply minimize $\left(\frac{4}{3}\right)^{\frac{\alpha}{2}} k^{\frac{1}{\alpha}}$, as this will already give the right asymptotic scaling. Since the function diverges to infinity when $\alpha$ tends to zero or to infinity, we find the minimum by setting the derivative of $\frac{\alpha}{2}\log\left(\frac{4}{3}\right) + \frac{1}{\alpha}\log k$ to zero:

$$\frac{1}{2}\log\left(\frac{4}{3}\right) - \frac{1}{\alpha^2}\log k = 0 \iff \alpha = \alpha^* = \frac{\sqrt{2}}{\log^{1/2}(4/3)}\log^{1/2}(k).$$

Taking $\alpha = \lfloor\alpha^*\rfloor$, we obtain

$$\text{bond}(\text{GHZ}_k) \leq \frac{8}{3}\frac{\sqrt{2}}{\log^{1/2}(4/3)}\gamma k^{\frac{1}{2}+\frac{\sqrt{2}\log^{1/2}(4/3)}{\log^{1/2}k}+\frac{\log\log k}{2\log k}}. \tag{34}$$

Since $\log^{1/2}(k) \geq \log\log k$, we can find $c$ positive as claimed in (28). We can improve this bound by minimizing the right hand side of (33) instead, obtaining

$$\text{bond}(\text{GHZ}_k) \leq \frac{8\gamma}{3\log(4/3)} k^{\frac{1}{2}+\frac{\sqrt{1+2\log\left(\frac{4}{3}\right)\log(k)}}{\log(k)}+\frac{\log\left(-1+\sqrt{1+2\log\left(\frac{4}{3}\right)\log(k)}\right)}{\log(k)}}$$

$$= \frac{8\gamma}{3\log(4/3)} k^{\frac{1}{2}} e^{\sqrt{1+2\log(4/3)\log(k)}}\left(-1 + \sqrt{1+2\log(4/3)\log(k)}\right).$$

Note that the asymptotic scaling of this bound is the same as the one we had obtained by minimizing $\left(\frac{4}{3}\right)^{\frac{\alpha}{2}} k^{\frac{1}{\alpha}}$, as we claimed.

To get the second part of (29), let instead

$$m = \alpha n^{\alpha+2} \leq \alpha\left(\frac{4}{3}\right)^{1+\frac{\alpha}{2}} k^{\frac{\alpha+2}{2\alpha}},$$

so that $k \geq \left(\frac{4}{3}\right)^{-\alpha}\left(\frac{m}{\alpha}\right)^{\frac{2\alpha}{\alpha+2}}$. Then (32) implies for any $\alpha \geq 1$ that

$$\text{MaMu}_{2\gamma m} \geq \text{GHZ}_{\left(\frac{4}{3}\right)^{-\alpha}\left(\frac{m}{\alpha}\right)^{\frac{2\alpha}{\alpha+2}}}.$$

Ideally, we would like to take the maximum over $\alpha$ to obtain the best lower bound. Instead, we decide here to maximize the easier function

$$-\alpha\log(4/3) + \frac{2\alpha}{\alpha+2}\log m,$$

neglecting the additional summand depending on $-\log\alpha$, since this will already be sufficient to get the desired scaling. The maximum is attained at $\alpha$ satisfying

$$-\log(4/3) + 4\frac{\log m}{(\alpha+2)^2} = 0 \iff \alpha = \alpha^{**} = 2\frac{\log^{1/2} m}{\log^{1/2}(4/3)} - 2,$$

again since the function is smaller or equal to zero for $\alpha$ equal to zero or tending to infinity. Since both $(4/3)^{-\alpha}$ and $(m/\alpha)^{\frac{2\alpha}{\alpha+2}}$ are decreasing in $\alpha$, substituting $\alpha = \lfloor\alpha^{**}\rfloor$, we obtain that

$$\left(\frac{4}{3}\right)^{-\alpha} \leq \left(\frac{4}{3}\right)^{2} (m^2)^{-\frac{\log^{1/2}(4/3)}{\log^{1/2}(m)}},$$

and

$$\left(\frac{m}{\alpha}\right)^{\frac{2\alpha}{\alpha+2}} \le (m^2)^{\left(1-\frac{\log^{1/2}(4/3)}{\log^{1/2}(m)}\right)\left[1+\frac{1}{\log m}\left(\log(2)-\frac{1}{2}\log\log(4/3)+\frac{1}{2}\log\log m\right)\right]},$$

which implies

$$\text{MaMu}_{2\gamma m} \ge \text{GHZ}_q \quad \text{with} \quad q = c\,(m^2)^{1-2\frac{\log^{1/2}(4/3)}{\log^{1/2} m}}, \tag{35}$$

for some positive constant $c$ which gives the scaling claimed in (29). $\qquad\square$

## 5.2 From $\text{MaMu}_k(m)$ to $\text{GHZ}_k(m)$: the general case

In this section, we will outline a method to obtain explicit degenerations from $\text{MaMu}_k(m)$ to $\text{GHZ}_k(m)$, generalizing some of the results for the 3-party case from the previous section to $m$ parties. We will then work out in detail the case for $m = 4$ as an example. We leave the reverse bound as an open problem.

As before, $\text{MaMu}_k(m)$ will denote a network of $m$ parties arranged on a circle each sharing a maximally entangled state with $k$ levels with each of its two nearest neighbours. The goal is to find a local linear transformation $A_l(\varepsilon)$ at each vertex $l$ depending polynomially on $\varepsilon$ such that the leading contribution in $\varepsilon$ of the resulting state is an $m$-party GHZ state with $k'$ levels

$$\left(\bigotimes_{l=1}^m A_l(\epsilon)\right) \sum_{i_1,\ldots,i_m=1}^k |i_1 i_2\rangle |i_2 i_3\rangle \ldots |i_m i_1\rangle = \epsilon^d\ \text{GHZ}_{k'}(m) + \mathcal{O}\left(\epsilon^{d+1}\right),$$

where $k'$ should be as large as possible and the kets indicate the grouping of parties. Following [43], we choose the operators $A_l(\epsilon)$ diagonal in the local product basis, i.e. $A_l(\varepsilon)|i,j\rangle = \varepsilon^{p_l(i,j)}|i,j\rangle$. In addition, we require that the leading order contribution in $\varepsilon$ is given by those vectors $|i_1, i_2\rangle \cdots |i_m, i_1\rangle$, that satisfy a certain system of linear equations, i.e. $\sum_l c_l i_l = g$ with coefficients vector $c_l$ and inhomogeneity $g$ belonging to $\mathbb{Z}^\nu$ for some integer $\nu$ [43]. This last condition is equivalent to the requirement that the vector $\sum_l c_l i_l - g$ is the zero vector, which can be reexpressed by the norm condition

$$0 = \left\langle \sum_l c_l i_l - g \,\middle|\, \sum_l c_l i_l - g \right\rangle = \sum_{l=0}^m \left(\langle c_l|c_l\rangle i_l^2 - (\langle g|c_l\rangle + \langle c_l|g\rangle)i_l\right) + \langle g|g\rangle + \sum_{l\neq l'} \langle c_l|c_{l'}\rangle i_l i_{l'}. \tag{36}$$

However, we have to connect this expression back to the local operations $A_l(\varepsilon)$. Indeed, we have to ensure that (36) can be generated by a product of local degenerations of the form

$$A_l(\epsilon)|ij\rangle = \epsilon^{p_l(i,j)}|ij\rangle,$$

namely $\sum_l p_l(i_l, i_{l+1}) = d + \left\|\sum_l c_l i_l - g\right\|_2^2$, which can always be achieved for all the terms in (36) that depend at most on a single index $l$. However, for the cross-terms this requires $\langle c_l|c_{l'}\rangle = 0$ if $|l - l'| > 1$, forcing the vectors $c_l$ into an orthogonal representation of the cycle graph (giving a lower bound on $\nu$), in which case we obtain

$$\left(\bigotimes_{l=1}^m A_l(\epsilon)\right) \sum_{i_1,\ldots,i_m=1}^k |i_1 i_2\rangle |i_2 i_3\rangle \ldots |i_m i_1\rangle = \sum_{i_1,\ldots,i_m=1}^k \epsilon^{\left\langle \sum_l c_l i_l - g \middle| \sum_l c_l i_l - g \right\rangle} |i_1 i_2\rangle |i_2 i_3\rangle \ldots |i_m i_1\rangle. \tag{37}$$

Furthermore, we have to ensure that the leading contribution, given by

$$\sum_{\substack{i_1,\ldots i_m=1 \\ \sum_l c_l i_l = g}}^m |i_1, i_2\rangle \cdots |i_m, i_1\rangle \tag{38}$$

is indeed locally unitarily equivalent to a GHZ state, i.e. consists of an equal weight super-position of product states $\psi_r = \psi_{r,1} \otimes \cdots \otimes \psi_{r,m}$, such that $\langle \psi_{r,l} | \psi_{r',l} \rangle = \delta_{r,r'}$. Since (38) is a superposition of vectors of the form $|i_1, i_2\rangle \cdots |i_m, i_1\rangle$ this means that fixing a pair of indices $i_{l'}, i_{l'+1}$ at any vertex $l$ the linear equation $\sum_l c_l i_l = g$ must have at most one unique solution in the remaining $i_l$. One way of ensuring this is to choose the vectors $c_l, c_{l'}$ linearly independent, whenever $|l - l'| > 1$. In other words, we have to choose the vectors $(c_l)_l$ in such a way that if we remove any subset of vectors that share a vertex, the remaining ones have to be linearly independent. The maximal dimension of the GHZ state we can extract is then given by the number of integer solutions to the equation

$$\sum_{l=0}^{m} c_l i_l = g \,, \tag{39}$$

where we optimize over the inhomogeneity $g$. One can get a bound on the number of these solutions by a probabilistic argument with respect to the inhomogeneity $g$. However, in order to talk about the finite $m$ case, we are going write down an explicit expression for (39) that satisfies all the necessary properties, i.e. $\langle c_l | c_{l'} \rangle = 0$ for $l' \notin \{l-1, l, l+1\}$ and $\{c_l\}_{l=0}^{m} \setminus \{c_j, c_{j+1}\}$ linearly independent for all $j$. We define the equations inductively starting from the four-party case

$$\begin{pmatrix} 1 \\ 1 \end{pmatrix} i_1 + \begin{pmatrix} -1 \\ 0 \end{pmatrix} i_2 + \begin{pmatrix} 1 \\ -1 \end{pmatrix} i_3 + \begin{pmatrix} 0 \\ 1 \end{pmatrix} i_4 = g \,. \tag{40}$$

Now adding a new vertex and edge into the cycle between $i_4$ and $i_1$ means that now $c_4$ has to be orthogonal to $c_1$ and the new $c_5$ should be orthogonal to all vectors except $c_1$ and $c_4$. This can be achieved by the choice

$$\begin{pmatrix} 1 \\ 1 \\ 1 \end{pmatrix} i_1 + \begin{pmatrix} -1 \\ 0 \\ 0 \end{pmatrix} i_2 + \begin{pmatrix} 1 \\ -1 \\ 0 \end{pmatrix} i_3 + \begin{pmatrix} 0 \\ 1 \\ -1 \end{pmatrix} i_4 + \begin{pmatrix} 0 \\ 0 \\ 1 \end{pmatrix} i_5 = g \,. \tag{41}$$

This procedure can be repeated leading to the following linear system for the $k$-cycle

$$\begin{pmatrix} 1 & -1 & 1 & 0 & 0 & \cdots & 0 & 0 \\ 1 & 0 & -1 & 1 & 0 & & 0 & 0 \\ 1 & 0 & 0 & -1 & 1 & & 0 & \\ 1 & 0 & 0 & 0 & -1 & & & \\ \vdots & \vdots & \vdots & \vdots & \vdots & & \vdots & \vdots \\ 1 & 0 & 0 & 0 & 0 & & 1 & 0 \\ 1 & 0 & 0 & 0 & 0 & & -1 & 1 \end{pmatrix} \cdot \vec{i} = g \,. \tag{42}$$

In order to find the integer solutions to this problem, we employ the Smith normal form of the matrix on the left-hand side, which gives the general solution vector

$$\vec{i} = \begin{pmatrix} z_1 + z_2 + A_1 \\ (m-2)(z_1 - z_2) + z_2 + A_2 \\ (m-3)(z_1 - z_2) + z_2 \\ \vdots \\ 2(z_1 - z_2) + z_2 + A_{m-2} \\ z_1 \\ z_2 \end{pmatrix} \,, \tag{43}$$

where $z_1, z_2$ are arbitrary integers and the constants $(A_l)$ depend on the choice of $g$ by a simple linear integer transformation given by the Smith normal form. In order to obtain the relevant solutions for our specific problem, we have to impose the upper and lower bounds 0 and $k-1$ if the original maximally entangled states are of dimension $k$ for each entry of the solution vector $\vec{i}$.

**The case $m = 4$**   In the case $m = 4$, (43) leads to the inequalities

$$0 \leq \begin{pmatrix} z_1 + z_2 + g_2 \\ 2z_1 - z_2 + g_2 - g_1 \\ z_1 \\ z_2 \end{pmatrix} \leq n - 1 \,.$$

Choosing $g_2 = g_1 \in \{\frac{k}{2}, \frac{k-1}{2}\}$ depending on whether $k$ is even or odd leads to the lower bound on the number of solutions of the form $\frac{k^2+1}{2}$ for odd dimensions and $\frac{k^2}{2}$ for even $k$.

This shows $\mathrm{MaMu}_k(4) \trianglerighteq \mathrm{GHZ}_{\lceil \frac{k^2}{2} \rceil}(4)$, i.e. that we can locally degenerate from a cycle of four maximally entangled states with $k$ levels to a four party GHZ state of $\lceil \frac{k^2}{2} \rceil$ levels. Hence on the level of plaquette states, we can degenerate from pairwise maximally entangled states on four parties with $\lceil \sqrt{2D} \rceil$ levels to a GHZ state on four parties of $D$ levels. Taking into account that in a two-dimensional square lattice the bond dimension of neighbouring plaquette states have to be combined (see Figure 2 (c) and (e)), this means that semi-injective PEPS on the two-dimensional square lattice based on GHZ states as introduced in [27] with bond dimension $D$ can be represented as a normal PEPS of bond dimension $2D$. By our theorem, expectation values for these generalized PEPS can hence be computed from expectation values of normal PEPS, for which highly optimized numerical codes exist.

## 5.3   Bond dimension of $\lambda$ is strictly larger than 2

As discussed in Section 3, in [32] the PEPS representation of the RVB-state is obtained via the multipartite entangled state

$$\lambda = \sum_{i,j,k=0}^{2} \varepsilon_{i,j,k} \, |i, j, k\rangle + |2, 2, 2\rangle = \triangleright \,, \tag{44}$$

with $\varepsilon$ denoting the completely antisymmetric tensor such that $\epsilon_{0,1,2} = 1$. In [32], the state $\lambda$ was obtained as a restriction from $_3\triangleright_3^3$, obtaining the same PEPS representation of the RVB state with bond dimension 3 from [31]. It turns out that is sub-optimal: the tensor $\lambda$ can be obtained also as a restriction from $_3\triangleright_2^2$, using the following MPS representation:

$$M_0^{[1]} = \frac{1}{2}\begin{pmatrix} 0 & 1 & 0 \\ 1 & 0 & 0 \end{pmatrix} \qquad M_1^{[1]} = \begin{pmatrix} 0 & -1 & 0 \\ 1 & 0 & 0 \end{pmatrix} \qquad M_2^{[1]} = \begin{pmatrix} 1 & 0 & 1 \\ 0 & -1 & 0 \end{pmatrix} \tag{45}$$

$$M_0^{[2]} = \frac{1}{2}\begin{pmatrix} 0 & 1 \\ 1 & 0 \\ 0 & 0 \end{pmatrix} \qquad M_1^{[2]} = \begin{pmatrix} 0 & -1 \\ 1 & 0 \\ 0 & 0 \end{pmatrix} \qquad M_2^{[2]} = \begin{pmatrix} 1 & 0 \\ 0 & -1 \\ 1 & 0 \end{pmatrix} \tag{46}$$

$$M_0^{[3]} = \frac{1}{2}\begin{pmatrix} 0 & 1 \\ 1 & 0 \end{pmatrix} \qquad M_1^{[3]} = \begin{pmatrix} 0 & -1 \\ 1 & 0 \end{pmatrix} \qquad M_2^{[3]} = \begin{pmatrix} 1 & 0 \\ 0 & -1 \end{pmatrix}. \tag{47}$$

This leads to a PEPS representation of the RVB state where the bond dimension is reduced from 3 to 2 on two of the edges of each triangle of the kagome lattice.

We now prove that this representation is optimal, i.e. that $\lambda$ cannot be represented as an MPS of bond dimension 2. This shows in particular a separation of bond and border bond PEPS representations on the kagome lattice, as the PEPS representation for $\lambda$ obtained in (20) has border bond dimension equal to (2,2,2).

**Proposition 19.** $2 \triangleright_2^2 \not\equiv \triangleright$

*Proof.* Given the general form of an MPS, we have to show that there exist no triples of $2 \times 2$-matrices $(A_i)$, $(B_j)$, $(C_k)$ satisfying

$$\text{tr}(A_i B_j C_k) = \varepsilon_{i,j,k} + \delta_{2,i,j,k}.\tag{48}$$

We first note, that the trace on the left hand side gives rise to the usual MPS gauge freedom, were we can substitute $A_i \mapsto X A_i Y$, $B_j \mapsto Y^{-1} B_j Z$ and $C_k \mapsto Z^{-1} C_k X^{-1}$ for $X, Y, Z \in GL(3)$. Next, we observe that the antisymmetric part of $\lambda$ is invariant under $M \otimes M \otimes M$ with $M \in SL(3)$, the special linear group. Hence, restricting to matrices of the form $M = R \oplus |2\rangle\langle 2|$, with $R \in SL(2)$, which in addition leave $|2,2,2\rangle$ invariant, we also have $(M \otimes M \otimes M)\lambda = \lambda$. Thus taking this physical symmetry plus the $Y, Z$ gauge transformation together and restricting for the moment to the $2 \times 2 \times 2$ tensor $\widetilde{B} = (B_0, B_1)$, we see that we can apply any operator $K_1 \otimes K_2 \otimes K_3$ with $K_i \in GL(2)$ to $\widetilde{B}$ without changing (48) if we transform $(A_i)_i$ and $(C_k)_k$ accordingly. However $GL(2)^3$ orbits of $2 \times 2 \times 2$-tensors are known explicitly [21], and we can use this freedom in order to reduce $B_0$ and $B_1$ to seven different normal forms, for which we have to obtain a contradiction. In addition to the null tensor and the product state, these seven classes encompass the bipartite entanglement between only two parties, the W state and the GHZ state. We will now go through all the cases.

**null tensor** In this case, both $B_0$ and $B_1$ are equal to the zero matrix, which leads for example to $\text{tr}(A_i B_1 C_k) = 0$, which clearly contradicts (48).

**product state** In this case, $\widetilde{B}$ can be chosen as $|0\rangle |0\rangle |0\rangle$, which implies $B_0 = |0\rangle\langle 0|$ and $B_1$ equal to the zero matrix. Hence, $\text{tr}(A_i B_1 C_k) = 0$ for all $i, k$ leads to the same contradiction as for the null tensor.

**bipartite entanglement (3 cases)** Depending on the two tensor factors that share the maximally entangled state, $\widetilde{B}$ can be chosen as $|000\rangle + |011\rangle$, $|000\rangle + |101\rangle$ or $|000\rangle + |110\rangle$. In the first case $B_0 = \mathbb{1}$ and $B_1 = 0$, which brings us back to the previous situation. In the remaining two cases $B_0 = |0\rangle\langle 0|$ and $B_1 = |0\rangle\langle 1|$ or $B_1 = |1\rangle\langle 0|$, respectively.

**GHZ state** In this case, $\widetilde{B} = |000\rangle + |111\rangle$ leading to $B_0 = |0\rangle\langle 0|$ and $B_1 = |1\rangle\langle 1|$.

**W state** Finally, in this case $\widetilde{B}$ can be chosen as $|000\rangle + |101\rangle + |110\rangle$, giving $B_0 = |0\rangle\langle 0|$ and $B_1 = |0\rangle\langle 1| + |0\rangle\langle 1|$.

In all the cases which we have not immediately discarded, we see that $B_0$ can be chosen as $|0\rangle\langle 0|$ while $B_1$ can either be $|1\rangle\langle 1|$, $|1\rangle\langle 0|$, $|0\rangle\langle 1|$ or $|0\rangle\langle 1| + |1\rangle\langle 0|$. We now want to show that neither of these cases are possible. We start by decomposing the matrices $A_i$ and $C_k$ as

$$A_i = |a_i\rangle\langle 0| + |\tilde{a}_i\rangle\langle 1|, \quad C_k = |0\rangle\langle c_k| + |1\rangle\langle \tilde{c}_k|$$

for vectors $|a_i\rangle, |\tilde{a}_i\rangle, |c_k\rangle, |\tilde{c}_k\rangle \in \mathbb{C}^2$. Since we have reduced the problem to the case $B_0 = |0\rangle\langle 0|$, we have that

$$\text{tr}(A_i B_0 C_k) = \langle c_k | a_i \rangle = \epsilon_{i,0,k}.$$

In particular, we have that $\langle c_1|a_2\rangle = 1$, $\langle c_2|a_1\rangle = -1$, implying that none of these vectors can be the zero vector. Together with $\langle c_2|a_2\rangle = 0$ this means that $\mathrm{span}\{|a_1\rangle, |a_2\rangle\} = \mathbb{C}^2$, and thus necessarily $|c_0\rangle$ has to be 0, since the trace condition forces it to be orthogonal to both $a_1$ and $a_2$. Similarly, we have that $\mathrm{span}\{|c_1\rangle |c_2\rangle\} = \mathbb{C}^2$ and that $|a_0\rangle = 0$.

Let us denote the matrix entries of $B_2$ as $b_{i,j} = \mathrm{tr}(B_2 |j\rangle\langle i|)$ for $i, j = 0, 1$, and let us consider the vectors

$$\left|a_i'\right\rangle = b_{0,1}\left|a_i\right\rangle + b_{1,1}\left|\tilde{a}_i\right\rangle, \quad \left|c_k'\right\rangle = \overline{b_{1,0}}\left|c_k\right\rangle + \overline{b_{1,1}}\left|\tilde{c}_k\right\rangle.$$

Then it holds that

$$\left\langle c_k'\middle|a_i'\right\rangle = b_{1,1}\,\mathrm{tr}(A_i B_2 C_k) + (b_{1,0}b_{0,1} - b_{0,0}b_{1,1})\langle c_k|a_i\rangle = b_{1,1}(\epsilon_{i,2,k} + \delta_{2,i,k}) - \det(B_2)\epsilon_{i,0,k}.$$

In particular $\left\langle c_k'\middle|a_i'\right\rangle = 0$ for $(i, k) = \{(0,0), (0,2), (2,0)\}$. Therefore, they define an orthogonal representation of $\mathfrak{K}_{2,2}^0$: by Lemma 18, either $\left\langle c_2'\middle|a_2'\right\rangle = 0$, or either $\left|a_0'\right\rangle$ or $\left|c_0'\right\rangle$ is zero. We can exclude the latter case, since this would imply that either $A_0$ or $C_0$ is zero, which we already know leads to a contradiction. Therefore $\left\langle c_2'\middle|a_2'\right\rangle = b_{1,1} = 0$. In the same way, defining

$$\left|a_i''\right\rangle = b_{0,1}\left|a_i\right\rangle + b_{0,0}\left|\tilde{a}_i\right\rangle, \quad \left|c_k''\right\rangle = \overline{b_{1,0}}\left|c_k\right\rangle + \overline{b_{0,0}}\left|\tilde{c}_k\right\rangle,$$

it holds that

$$\left\langle c_k''\middle|a_i''\right\rangle = b_{0,0}(\epsilon_{i,2,k} + \delta_{2,i,k}) - \det(B_2)\epsilon_{i,0,k},$$

so we can conclude that also $b_{0,0} = 0$.

We will now consider the four possibilities we have for $B_1$, driving each one of them to a contradiction, and therefore showing that no MPS representation of $\lambda$ with bond dimension 2 is possible.

1. $B_1 = |1\rangle\langle 0|$: We get a contradiction since $\mathrm{tr}(A_2 B_1 C_0)$ should be $-1$, but $B_1 C_0 = 0$.

2. $B_1 = |0\rangle\langle 1|$: We get a contradiction since $\mathrm{tr}(A_0 B_1 C_2)$ should be 1, but $A_0 B_1 = 0$.

3. $B_1 = |0\rangle\langle 1| + |1\rangle\langle 0|$: In this case, $\mathrm{tr}(A_i B_1 C_k) = \langle \tilde{c}_k|a_i\rangle + \langle c_k|\tilde{a}_i\rangle = \epsilon_{i,1,k}$, and in particular $\mathrm{tr}(A_1 B_1 C_0) = \langle \tilde{c}_0|a_1\rangle$ since $|c_0\rangle = 0$. From this equation it follows that

$$\mathrm{tr}(A_1 B_2 C_0) = b_{0,1}\langle \tilde{c}_0|a_1\rangle + b_{1,1}\langle c_0|\tilde{a}_1\rangle = b_{0,1}\,\mathrm{tr}(A_1 B_1 C_0) = 0 \neq 1,$$

so we obtain a contradiction.

4. $B_1 = |1\rangle\langle 1|$: We see that $\mathrm{tr}(A_i B_1 C_k) = \langle \tilde{c}_k|\tilde{a}_i\rangle = \epsilon_{i,1,k}$, so reasoning in the same way as before we see that $|\tilde{a}_1\rangle = |\tilde{c}_1\rangle = 0$ and that $|\tilde{a}_0\rangle, |\tilde{a}_2\rangle, |\tilde{c}_0\rangle$ and $|\tilde{c}_2\rangle$ are non-zero, therefore reducing to the case where

$$
\begin{aligned}
A_0 &= |\tilde{a}_0\rangle\langle 1|, & C_0 &= |1\rangle\langle \tilde{c}_0|, \\
A_1 &= |a_1\rangle\langle 0|, & C_1 &= |0\rangle\langle c_1|, \\
A_2 &= |a_2\rangle\langle 0| + |\tilde{a}_2\rangle\langle 1|, & C_2 &= |0\rangle\langle c_2| + |1\rangle\langle \tilde{c}_2|.
\end{aligned}
$$

Considering

$$\mathrm{tr}(A_1 B_2 C_0) = \langle \tilde{c}_0|a_1\rangle b_{0,1} = 1, \quad \mathrm{tr}(A_0 B_2 C_1) = \langle c_1|\tilde{a}_0\rangle b_{1,0} = -1,$$

we obtain that $b_{0,1}, b_{1,0}$ and $\langle \tilde{c}_0|a_1\rangle, \langle c_1|\tilde{a}_0\rangle$ are non-zero. On the other hand since $b_{1,1} = 0$ we have that

$$0 = \mathrm{tr}(A_2 B_2 C_0) = \langle \tilde{c}_0|a_2\rangle b_{0,1}, \quad 0 = \mathrm{tr}(A_0 B_2 C_2) = -\langle c_2|\tilde{a}_0\rangle b_{1,0},$$

and since $b_{0,1} \neq 0$ and $b_{1,0} \neq 0$ we see that necessarily $\langle \tilde{c}_0 | a_2 \rangle = \langle c_2 | \tilde{a}_0 \rangle = 0$. Therefore $|a_2\rangle$ is proportional to $|\tilde{a}_0\rangle$ and similarly $|c_2\rangle$ is proportional to $|\tilde{c}_0\rangle$, and so it follows that

$$
\begin{aligned}
\frac{\langle \tilde{c}_2 | a_2 \rangle}{\langle c_2 | \tilde{a}_2 \rangle} &= \frac{\langle \tilde{c}_2 | a_2 \rangle}{\langle c_2 | \tilde{a}_2 \rangle} \cdot \frac{\langle c_1 | \tilde{a}_0 \rangle}{\langle c_1 | \tilde{a}_0 \rangle} \cdot \frac{\langle \tilde{c}_0 | a_1 \rangle}{\langle \tilde{c}_0 | a_1 \rangle} \\
&= \frac{\langle \tilde{c}_2 | \tilde{a}_0 \rangle}{\langle c_2 | a_1 \rangle} \cdot \frac{\langle c_1 | a_2 \rangle}{\langle c_1 | \tilde{a}_0 \rangle} \cdot \frac{\langle \tilde{c}_0 | a_1 \rangle}{\langle \tilde{c}_0 | \tilde{a}_2 \rangle} = \frac{1}{-1} \cdot \frac{1}{\langle c_1 | \tilde{a}_0 \rangle} \cdot \frac{\langle \tilde{c}_0 | a_1 \rangle}{-1} = -\frac{b_{1,0}}{b_{0,1}}.
\end{aligned}
$$

This leads to a contradiction since

$$
\mathrm{tr}(A_2 B_2 C_2) = b_{0,1} \langle \tilde{c}_2 | a_2 \rangle + b_{1,0} \langle c_2 | \tilde{a}_2 \rangle = 0 \neq 1.
$$

$\square$

# 6 Conclusions

We have shown that analyzing the geometry of entangled states and transformations between entanglement structures provides a framework for the construction of more efficient tensor network representations. Starting from local improvements on the level of plaquette states, we obtain optimized tensor network representations on the entire lattice. We provide two methods to construct such local improvements: restrictions and degenerations.

Using geometrical tools, our main result allows us to lift the local approximate conversion originating from degenerations on the level of plaquette states to an exact representation of the tensor network state on the entire lattice, given as a superposition of tensor network states with smaller bond dimension, the number of which scales linearly in the system size. In addition, our general result gives a prescription of how to leverage this bond dimension reduction in order to reduce the computational cost of computing expectation values. More precisely, we describe a parallel contraction algorithm to compute physical expectation values $\langle T, OT \rangle$ of the original state as $\sum_i^{2eF} \gamma_i \langle V_i, OV_i \rangle$, where each $V_i$ is given as PEPS of lower bond dimension than $T$.

As an example of application of these techniques, we studied explicitly the RVB state on the kagome lattice. We present two improvements on the representation from [32]. The first is obtained by considering bonds of different dimensions, allowing us to arrive at the optimal representation, where two out of three bonds on each triangle can be reduced to bond dimension 2 instead of 3. This leads to saving in the cost of computing contractions (which for the sake of completeness we detailed in Appendix A). The second improved representation is obtained by considering the more general case of degenerations from the plaquette state  : we can then find a border bond dimension 2 representation of the RVB state, which again is optimal in terms of this effective bond dimension.

More generally, given an entanglement structure $\Phi$ built from locally distributed multipartite entangled states, our result allows to characterize the variational class given by the set of states obtained by applying local maps $\{A_i(\epsilon)\}_{i=0}^L$ which are polynomial of degree $e$ in $\epsilon$, and then taking the limit $\epsilon$ to zero. Each state obtained in this fashion is specified by a polynomial number of parameters. Our main theorem then shows that this gives us access to states which arise as a superposition of a linear number of states represented by $\Phi$, going beyond the states representable by a single tensor network state of this bond dimension. Nevertheless, their expectation values can be efficiently computed by interpolation.

An interesting question that we leave open for future research is how to optimize efficiently within the set of tensor network states that arise as degenerations, e. g. in order to minimize the energy of a local Hamiltonian. Since degenerations are still given by a local tensor albeit

depending polynomially on a free parameter, we expect that a local optimization step along the lines of the known tensor network techniques will be possible. However, one has to carefully take into account the additional constraint for obtaining an honest degeneration and we leave the details of such an optimization procedure for future work.

## Acknowledgments

We thank Ignacio Cirac, Christian Krumnow and David Pérez-García for helpful discussions. M. C. acknowledges the hospitality of the Center for Theoretical Physics at MIT, where part of this work was carried out.

**Funding information**    We acknowledge financial support from the European Research Council (ERC Grant Agreement no. 337603 and ERC Grant agreement no. 818761), the Danish Council for Independent Research (Sapere Aude) and VILLUM FONDEN via the QMATH Centre of Excellence (Grant no. 10059). A. L. acknowledges support from the Walter Burke Institute for Theoretical Physics in the form of the Sherman Fairchild Fellowship as well as support from the Institute for Quantum Information and Matter (IQIM), an NSF Physics Frontiers Center (NFS Grant PHY-1733907). A. H. W. thanks the VILLUM FONDEN for its support with a Villum Young Investigator Grant (Grant No. 25452) and the Humboldt Foundation for its support with a Feodor Lynen Fellowship. P. V. acknowledges support by the National Research, Development and Innovation Fund of Hungary within the Quantum Technology National Excellence Program (Project Nr. 2017-1.2.1-NKP-2017-00001) and via the research grants K124152, KH129601.

## A    Computational complexity of tensor-contractions

In this appendix, we derive estimates on the computational cost of exactly and approximately contracting PEPS networks for the two-dimensional square lattice and the kagome lattice. We will subsequently discuss a specialized contraction strategy for the RVB state from the literature.

### A.1    Exact contraction of the RVB state on the kagome lattice

Before discussing contractions of tensor networks with varying bond dimension in the subsequent subsection, let us briefly comment on the computational complexity of exactly contracting the RVB state on the kagome lattice in regards to bond and border bond PEPS representations. One strategy employed to contract a PEPS on a lattice is to treat one boundary of the two PEPS layers as an MPS and to view the contraction of the remaining rows of PEPS tensors as the application of matrix product operators to this boundary MPS (see Figure 5 and 6). For a PEPS with bond dimension $D$ on the kagome lattice the computational cost of the contraction of a single local tensor into the boundary MPS is given by $\mathcal{O}(\chi^3 D^4) + \mathcal{O}(\chi^2 D^6 d)$ [45], with $d$ the physical dimension and $\chi$ denoting the bond dimension of the boundary MPS. However, since we do not reduce the bond dimension of the boundary MPS $\chi$, we can omit the final SVD, and the relevant scaling is simply $\mathcal{O}(\chi^2 D^6 d)$.

Contracting one full row of local PEPS tensors into the boundary MPS increases its bond dimension by a factor of $D^2$ due to the double layer structure of the network, i.e. $\chi_{i+1} = D^2 \chi_i$. In the case of an exact contraction, this bond dimension is not compressed after each step, and hence $\chi$ grows exponentially with $D^2$. Accordingly, if we consider the computational cost of computing an expectation value of a PEPS with bond dimension $D$ on a $2(L+1) \times 2(L+1)$

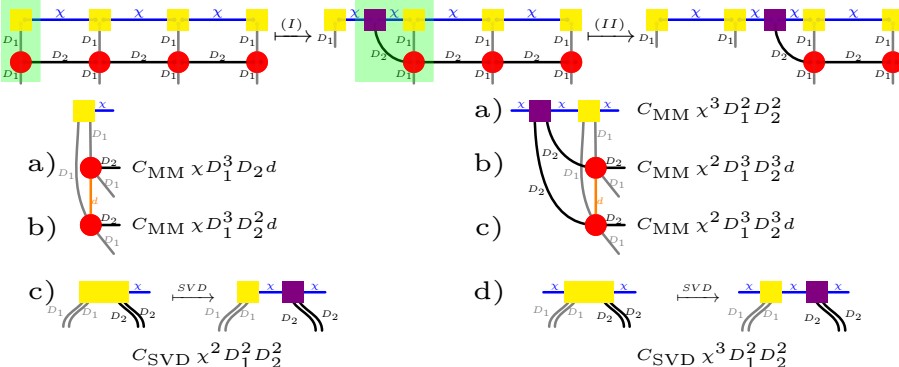

Figure 5: Approximate contraction of a PEPS network on the two-dimensional square lattice with the boundary-MPS method. The first row shows the initial step at the boundary (I) and the bulk-step (II), which is repeated until the right boundary of the network is reached. For simplicity, only a single layer of the two-layer PEPS-network is shown here, but each red circle in the upper row represents the two local PEPS tensors that have to be contracted along the invisible physical dimension. In the second row the detailed contractions of both PEPS-layers that are carried out in each step are depicted with their corresponding computational cost. Lines that terminate in a tensor at a given sub-step ( a)-d)) in a tensor represent the contractions carried out at this point, whereas lines not connected to a tensor at that level correspond to free indices. In total, the scaling is given by $(C_{\mathrm{MM}} + C_{\mathrm{SVD}})\chi^3 D_1^2 D_2^2 + 2C_{\mathrm{MM}}\chi^2 D_1^3 D_2^3 d$.

lattice and taking into account that we can use a boundary MPS from both sides of the lattice, the overall computational cost is given by

$$2(L+1)\sum_{l=1}^{L}\mathcal{O}(\chi_l^2 D^6 d) = \sum_{l=1}^{L}\mathcal{O}(\chi_0^2 (D^{2l})^2 D^6 d) = 2(L+1)\mathcal{O}(\frac{D^{4L}-1}{D^4-1}D^{10}d)\,, \tag{49}$$

with $\chi_0$ the bond dimension of the first boundary MPS. In case of a border PEPS representation with border bond dimension $D$ and error degree $eL$, i.e. linear in the system size as happens for the case of local degenerations of the plaquette tensor, Theorem 14 implies that we have to contract $2eL + 1$ PEPS in order to compute an exact expectation value, leading to a scaling of

$$2(L+1)(2eL+1)\mathcal{O}(\frac{D^{4L}-1}{D^4-1}D^{10}d)\,. \tag{50}$$

Accordingly, considering the bond dimension 3 PEPS representation of the RVB state in comparison to the border bond dimension 2 representation, we obtain a leading scaling of $\mathcal{O}(L\,3^{4L})$ versus a scaling of $\mathcal{O}(L^2 2^{4L})$, which gives an exponential improvement.

## A.2  PEPS with varying bond dimension

We now turn to the contraction of PEPS networks on the kagome and square lattice. In contrast to the results commonly stated in the literature, we will explicitly deal with the case of non-equal bond dimensions with respect to different virtual degrees of freedom and in the case of the kagome lattice also take into account different distributions of the legs in the two layers of the network. In all cases, we consider a boundary-MPS approach, where the PEPS tensors at the boundary of the network are considered as an MPS of fixed bond dimension $\chi$ to which the internal PEPS tensor regarded as MPOs are applied subsequently. All bounds are based on

the estimates $C_{MM}D_1D_2D_3$ for the computation of the product of two rectangular matrices of dimensions $D_1{\times}D_2$ and $D_2{\times}D_3$ and $C_{\text{SVD}}\chi D_1 D_2$ for the truncated singular value decomposition (SVD) of a $D_1{\times}D_2$ matrix to its largest $\chi$ singular values [46] with $C_{\text{MM}}$ and $C_{\text{SVD}}$ constants.

**Two-dimensional square lattice** Starting from one boundary of the lattice, the next row of the double of the contraction are depicted in Figure 5. Starting from the left-boundary, the first MPO-tensor (red circle) of the next row is contracted into the boundary-MPS and its bond dimension subsequently reduced to $\chi$ via an SVD (step (I)). The cost of each step in this contraction is indicated in the second row of Figure 5. In each of the steps a), b) and c), the contractions performed in that step are indicated by lines that terminate in a tensor at that level, all other lines count as free indices. In step I.a) for example the only contraction performed is with respect to the gray line connecting the yellow square and the red circle, whereas the remaining lines (two gray, one black, one orange) are free indices. Hence, this contraction can be seen as a multiplication between a $\chi D_1{\times}D_1$ matrix (yellow square) and a $D_1{\times}D_1 D_2 d$ matrix (red circle) leading to an overall cost of $C_{\text{MM}}\cdot \chi D_1^3 D_2 d$. The two red circles correspond to the two layers of the PEPS network. Hence, the overall cost for contracting the MPO into the boundary-MPS at the boundary is given by

$$C_{\text{SVD}}\,\chi^2 D_1^2 D_2^2 + C_{\text{MM}}\,\chi D_1^3 (D_2^2 + D_2)d.$$

In step (II), the sub-steps b) to d) are basically the same one as the steps a) to c) in step (I), however we first have to take care of the violet tensor resulting from the SVD performed in sub-step I.c. The overall computational cost is then given by

$$(C_{\text{MM}} + C_{\text{SVD}})\,\chi^3 D_1^2 D_2^2 + 2C_{\text{MM}}\,\chi^2 D_1^3 D_2^3 d\,.$$

Because this cost upper bounds the contraction cost at the boundary, cost of contracting each MPO tensor into the boundary-MPS tensor can be upper bounded by

$$(C_{\text{MM}} + C_{\text{SVD}})\,\chi^3 D_1^2 D_2^2 + 2C_{\text{MM}}\,\chi^2 D_1^3 D_2^3 d\,, \tag{51}$$

which agrees with the estimate for uniform bond dimension $\mathcal{O}(\chi^2 D^6 d) + \mathcal{O}(\chi^3 D^4)$ found in the literature [2, 3, 47].

**Kagome lattice** The situation for the kagome lattice is very similar when compared to the square two-dimensional lattice except more care has to be taken about how to associate the local tensors to the boundary-MPS tensors. The procedure we adopt here is depicted in Figure 6. In order to make the procedure more transparent, we first split the boundary vertices at the tip of each triangle into two lattice sites, before we start the contraction procedure. Fixing the three bond dimensions in each triangle for the full lattice, we can nevertheless distinguish their distribution for upwards ($K_1$, $K_2$, $K_3$) and downwards ($D_1$, $D_2$, $D_3$) pointing triangles. In comparison to the square two-dimensional lattice, we have to distinguish three different contraction steps, depending on whether we are contracting a tensor on the top right (I), the top left (II) or in the middle (III) of a hexagon. These three steps are then repeated until the right boundary of the kagome lattice is reached.

Figure 7 (I) to (III) depicts the details of these three steps, breaking down every step into the explicit tensor-contractions performed and how expensive they are in terms of the dimension of the indices of the involved local tensors. In order to realize improved savings, we allow different distributions of the three bond dimensions in the two triangles for the upper and lower PEPS-layer, indicated by $D_i^\uparrow/D_i^\downarrow$ or $K_i^\uparrow/K_i^\downarrow$, respectively. Taking the maximum over the different computational costs in the three different contractions steps for $\chi^2$ and $\chi^3$

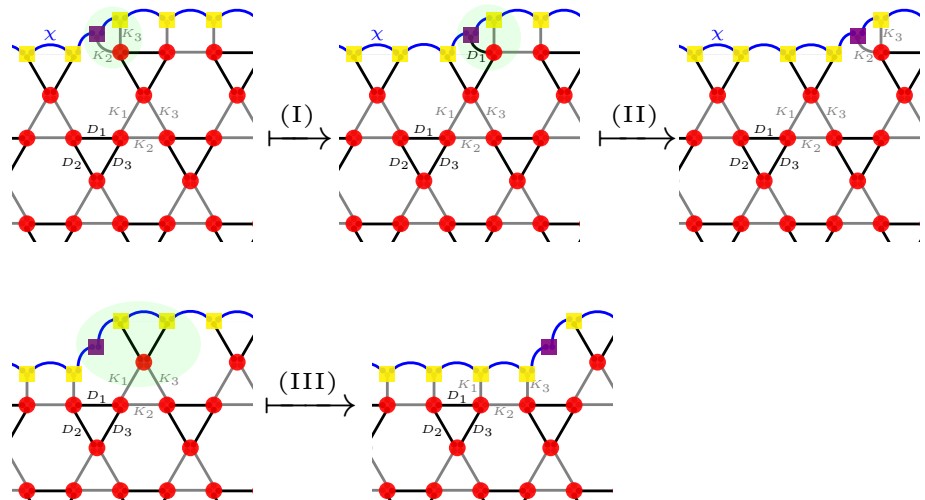

Figure 6: Approximate contraction of a PEPS network on the kagome lattice with the boundary-MPS method. Depending on the position of the local tensor to be contracted into the boundary-MPS in the kagome lattice, three different contractions have to be performed. We allow for different bond dimensions for up- ($K_i$) and downwards ($D_i$) pointing triangles.

separately, we can upper bound the computational cost of each of all local contractions by

$$
\begin{aligned}
&C_{\text{SVD}}\, \chi^3 \max(D_1^\uparrow D_1^\downarrow D_2^\uparrow D_2^\downarrow,\ D_3^\uparrow D_3^\downarrow K_2^\uparrow K_2^\downarrow,\ K_1^\uparrow K_1^\downarrow K_3^\uparrow K_3^\downarrow) \\
&+C_{\text{MM}}\, \chi^3 \max(K_2^\uparrow K_2^\downarrow K_3^\uparrow K_3^\downarrow,\ D_1^\uparrow D_1^\downarrow K_1^\uparrow K_1^\downarrow,\ D_2^\uparrow D_2^\downarrow D_3^\uparrow D_3^\downarrow) \\
&+C_{\text{MM}}\, \chi^2 d \max(K_2^\uparrow K_2^\downarrow K_3^\uparrow K_3^\downarrow D_1^\downarrow D_2^\uparrow + K_2^\downarrow K_3^\uparrow D_1^\uparrow D_1^\downarrow D_2^\uparrow D_2^\downarrow, \\
&\qquad\qquad D_1^\uparrow D_1^\downarrow K_1^\uparrow K_1^\downarrow D_3^\downarrow K_2^\uparrow + D_1^\downarrow K_1^\downarrow D_3^\downarrow D_3^\downarrow K_2^\uparrow K_2^\downarrow,\ D_2^\uparrow D_2^\downarrow D_3^\uparrow D_3^\downarrow K_1^\uparrow K_3^\uparrow + D_2^\downarrow D_3^\downarrow K_1^\uparrow K_1^\downarrow K_3^\uparrow K_3^\downarrow).
\end{aligned}
$$

In the case, where $D_1 = D_3 \leq D_2$, choosing the same distribution of the bond dimensions in both layers and both types of triangles, i.e. $D_i^\uparrow = D_i^\downarrow$ and $K_i = D_i$ we obtain an upper bound of

$$
(C_{\text{SVD}} + C_{\text{MM}}) \chi^3 D_1^2 D_2^2 + 2 C_{\text{MM}} \chi^2 D_1^3 D_2^3 d, \tag{52}
$$

which has a similar scaling as the square two-dimensional lattice. In the case where all bond dimensions are equal, we arrive at a scaling $\mathcal{O}(\chi^3 D^4) + \mathcal{O}(\chi^2 D^6 d)$ in correspondence with previous results in the literature [45].

## A.3 Approximate contraction of the RVB state

In this section, we apply the estimates on the computational costs of approximately contracting a PEPS on the kagome lattice derived so far in the context of the restrictions and degenerations for the RVB state. As discussed in Section 5.3 considering general restrictions and allowing for unequal bond dimension, we can obtain a PEPS representation of the RVB state, where two out of three bonds on each triangle of the kagome lattice are reduced to bond dimension 2 instead of 3 (see (45)).

According to (52), the computational cost of contracting a PEPS with bond dimensions satisfying $D_1 = D_3 \leq D_2$ around a triangular plaquette scales as $C_1 \chi^3 D_1^2 D_2^2 + C_2 \chi^2 D_1^3 D_2^3 d$, where $C_i$ are constants and $\chi$ denotes the bond dimension of the boundary-MPS. Hence, the optimized tensor network representation of the entanglement structure generated by ▷· underlying the RVB state reduces the prefactor of $\chi^3$ from $81 C_1$ to $36 C_1$ and for $\chi^2 d$ from $729 C_2$ to $216 C_2$ for restrictions. Note that this improvement applies to the contraction of all tensor

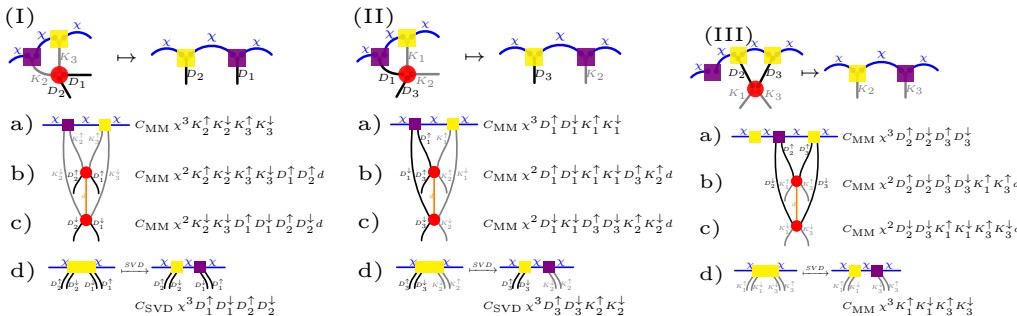

Figure 7: Illustration of the three contraction steps involved in the approximate contraction of a PEPS on the kagome lattice. (I) Details on step (I): The first row depicts the overall contraction step. (a)-(d) show the contractions performed in each sub-step and their corresponding computational cost. The superscript $\updownarrow$ indicates, whether a given index corresponds to the upper or lower level of the PEPS network. As in the case of the square two-dimensional lattice, lines terminating in a tensor for a given step are contracted, whereas non-terminated lines correspond to free indices of the tensors. (II) Details on step (II): The first row depicts the overall contraction step. (a)-(d) show the contractions performed in each sub-step and their corresponding computational cost. The superscript $\updownarrow$ indicates, whether a given index corresponds to the upper or lower level of the PEPS network. As in the case of the square two-dimensional lattice, lines terminating in a tensor for a given step are contracted, whereas non-terminated lines correspond to free indices of the tensors. (III) Details on step (III): The first row depicts the overall contraction step. (a)-(d) show the contractions performed in each sub-step and their corresponding computational cost. The superscript $\updownarrow$ indicates, whether a given index corresponds to the upper or lower level of the PEPS network. As in the case of the square two-dimensional lattice, lines terminating in a tensor for a given step are contracted, whereas non-terminated lines correspond to free indices of the tensors.

networks based on the $\triangleright\cdot$ entanglement structure on the kagome lattice. The same entanglement structure representing the RVB state is used in [32] to construct a family of quantum states which interpolates between the RVB state and a dimer state, which are believed to lie in different quantum phases. Since we have improved the PEPS representation of the entanglement structure behind all these states, the saving we have obtained for the RVB state applies to all of them. Note further that, obviously, there are ways to optimize the contraction cost for specific tensor networks. In [32], for instance, the kagome lattice is first transformed to a square lattice for which an RVB-specific improved double layer bond dimension is derived. For this particular example, our contraction scheme does not provide an advantage, however we would like to stress that our approach will allow a representation of border bond dimension 2 for any state obtained as a restriction from the $\Lambda$ entanglement structure.

If we consider the more general case of approximating the plaquette state $\triangleright\cdot$ in terms of degenerations, we can use the border bond dimension 2 representation of the RVB state ((20)). Employing the parallel contraction algorithm presented in Section 4, allows us to take advantage of this reduction also in the case of the RVB state or any other tensor network state based on the $\triangleright\cdot$ entanglement structure on the kagome lattice. The reduction to border bond dimension 2 reduces the prefactors for the computational effort for the contraction of each of the $4F$ expectation values $\langle V_i, OV_i \rangle$ to $16C_1$ for $\chi^3$ and $64C_2$ for $\chi^2 d$ as compared to $36C_1$ and $216C_2$ for the unbalanced optimal restriction with bond dimension $(2, 2, 3)$.

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
