# Peer review of "Tensor network representations from the geometry of entangled states"

_SciPost Physics, doi:SciPost Phys. 9, 042 (2020)_

## Round 2 · Referee Report · Anonymous (Referee 1) · 2020-7-9

Strengths

  1. The authors describe a relevant physics application for ideas from algebraic complexity theory, and show how theoretical advances in that field are relevant in the context of tensor network representations of quantum states.
  2. Some bounds on conversions and degenerations of multi-party states that are interesting in their own right.
  3. Very explicitly and carefully explained, working out an interesting example in full detail.

Weaknesses

  1. From this work it is not yet very clear (at least to this reviewer) how large the practical computational benefits could be in general (one explicit example is worked out in detail), and in what physical classes of states one may expect an advantage.
  2. In this work it is not explained how a variational method could be implemented that actually benefits from these representations. However, this has been addressed very recently by a subset of the authors in https://arxiv.org/abs/2006.16963.

Report

This is a very nice and original work. It works out the idea to consider states 'at the boundary' of the set of fixed bond dimension tensor network states. It translates concepts and results from the field of algebraic complexity to be relevant for tensor network states. A specific example is worked out rigorously. The paper is well written and satisfies all acceptance criteria. The results are fairly technical and, even though carefully explained, especially the plaquette conversion proofs may be hard to understand for the general physics reader. However, the possibility of introducing these ideas and techniques to a physics audience merits the publication in a physics journal.

Requested changes

  1. A few small typos: pg 10, line 2 "representable with by" -> "representable by", pg 14, last paragraph of section 4 "back to RVB state" -> "back to the RVB state", pg 19, below 3rd displayed equation a , should be omitted and logs -> log, pg 25 line 2 of section 6 "provide" -> "provides".
  2. The discussion in section 5.2 was hard to understand (at least to this reviewer). Perhaps the structure of the argument can be made a bit more clear?
  3. Something I wondered while reading is whether the construction in Theorem 14 is optimal in any sense, or whether for combining multiple plaquettes there could be better degenerations than the tensor product degeneration of single plaquette degenerations. If this question makes sense, perhaps the authors could make a remark on the possibilities for this.

---

## Round 2 · Referee Report · Anonymous (Referee 2) · 2020-7-10

Strengths

  1. nontrivial application of some of the iconic results of algebraic geometry to tensor networks
  2. paper is written in a very nice way, also understandable to people working on tensor networks but not experts in algebraic geometry

Weaknesses

  1. not so clear whether this leads to any practical gain in actual algorithms

Report

One of the more fascinating concepts in the mathematical treatment of tensors is the notion of border rank - for the purpose of this paper, the authors demonstrate that this means that certain tensors can be well approximated with tensors with a smaller rank. The rank of the tensors used in tensor network algorithms is the main factor in the cost of contracting them, so this is possibly a very valuable insight. The authors give some interesting examples, such as the description of RVB states on the Kagome lattice, which is certainly of interest to the tensor network community.

This is certainly a very interesting and original contribution to the field, and may potentially be very useful in the numerical treatment of tensor networks. However, I think that the main value of the paper is the fact that it opens up a dialogue between two different fields.

---

## Round 3 · Referee Report · Anonymous (Referee 1) · 2020-9-10

Report

Thanks, that is all good!

---

## Round 3 · Author Response

we would like to thank you and the reviewers for your input and comments to our manuscript.
Based on this feedback, we have uploaded a new version to the arxiv, which we hereby resubmit to SciPost.
A detailed list of changes is described below.

---

## Round 3 · List of Changes

Requested changes Report 1

  1. A few small typos: pg 10, line 2 "representable with by" -> "representable by", pg 14, last paragraph of section 4 "back to RVB state" -> "back to the RVB state", pg 19, below 3rd displayed equation a , should be omitted and logs -> log, pg 25 line 2 of section 6 "provide" -> "provides".

Thank you very much for spotting these, we have taken all of them into account.

  1. The discussion in section 5.2 was hard to understand (at least to this reviewer). Perhaps the structure of the argument can be made a bit more clear?

We have re-written part of this section, including more explanations about the method - in particular at the beginning of the section and hope that the argument is now clearer.

  1. Something I wondered while reading is whether the construction in Theorem 14 is optimal in any sense, or whether for combining multiple plaquettes there could be better degenerations than the tensor product degeneration of single plaquette degenerations. If this question makes sense, perhaps the authors could make a remark on the possibilities for this. Included comment after theorem

This is a very good observation and indeed it could be possible that better conversions can be obtained on larger patches. We have included a comment regarding this after the proof of Theorem 14 on page 13.

---

## Editorial Decision

published